# SCRIBES: WEB-SCALE SCRIPT-BASED SEMI-STRUCTURED DATA EXTRACTION WITH REINFORCEMENT LEARNING

**Shicheng Liu**[1]*, **Kai Sun**[2], **Lisheng Fu**[2], **Xilun Chen**[3], **Xinyuan Zhang**[2], **Zhaojiang Lin**[2], **Rulin Shao**[3,4], **Yue Liu**[2], **Anuj Kumar**[2], **Wen-tau Yih**[3], **Xin Luna Dong**[2]
[1] Stanford University,    [2] Meta Reality Labs,
[3] FAIR at Meta,    [4] University of Washington
shicheng@cs.stanford.edu, sunkaicn@meta.com

## ABSTRACT

Semi-structured content in HTML tables, lists, and infoboxes accounts for a substantial share of factual data on the web, yet the formatting complicates usage, and reliably extracting structured information from them remains challenging. Existing methods either lack generalization or are resource-intensive due to per-page LLM inference. In this paper, we introduce SCRIBES (**SCRI**pt-**B**ased Semi-Structured Content **E**xtraction at Web-**S**cale), a novel reinforcement learning framework that leverages layout similarity across webpages within the same site as a reward signal. Instead of processing each page individually, SCRIBES generates reusable extraction scripts that can be applied to groups of structurally similar webpages. Our approach further improves by iteratively training on synthetic annotations from in-the-wild CommonCrawl data. Experiments show that our approach outperforms strong baselines by over 13% in script quality and boosts downstream question answering accuracy by more than 4% for GPT-4o, enabling scalable and resource-efficient web information extraction.[1]

## 1 INTRODUCTION

A substantial volume of web data is stored in semi-structured formats such as HTML (Hyper-Text Markup Language) tables, lists, and infoboxes (Dong et al., 2014; Sun et al., 2025)[2]. Such content offers a rich source of factual information, yet its formatting complicates effective usage in downstream applications like question answering (Tan et al., 2025; Sun et al., 2025). Knowledge extraction aims to transform such data from raw HTML into structured representations (e.g., triples) (Wilks, 1997), but despite decades of research, this remains a major challenge at large scale. Existing approaches fall into two main categories. *Traditional information extraction (IE) methods*, such as wrapper induction (Kushmerick et al., 1997), graph mining (Crescenzi et al., 2001; Liu et al., 2003), layout-based methods (Zhai & Liu, 2005; Lockard et al., 2018), and Deep Neural Networks (Dalvi et al., 2011; Lockard et al., 2020), tend to be brittle and struggle to generalize over unseen data or schema. More recently, Large Language Model (LLM)-based methods have emerged that parse individual pages or construct Knowledge Graphs (KGs) using large models (Gutiérrez et al., 2024; Zhang & Soh, 2024; Ning et al., 2023; Chen & Bertozzi, 2023; Zhang et al., 2023; Bai et al., 2025). Although these methods can produce high-quality outputs, they are resource-intensive to apply at scale because they require invoking an LLM for every page.

*Can we extract knowledge from semi-structured content at the web scale both effectively and efficiently?* In this paper, we introduce **SCRIBES: SCRIpt-Based Semi-Structured Content Extraction at Web-Scale**, a novel approach for large-scale knowledge extraction. Given a webpage, SCRIBES leverages an LLM to generate an extraction script that applies to other pages within the same domain, which typically share highly similar layouts (Figure 2). Executing the script incurs

---

*Work done at Meta

[1]Code will be released at https://github.com/facebookresearch/SCRIBES.
[2]See Appendix B for a discussion of different types of webpages with semi-structured content.

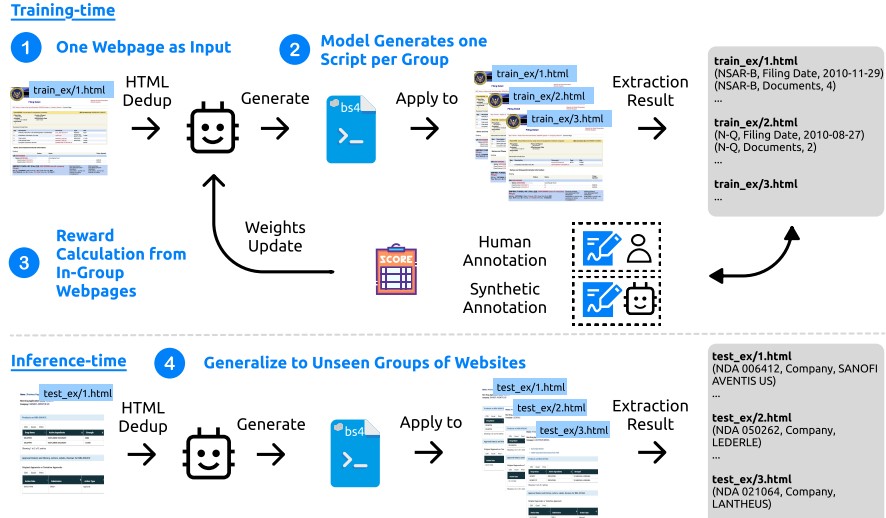

Figure 1: SCRIBES organizes similar webpages into groups under each website. During training, the model receives one representative webpage per group as input (pt. 1) and is tasked with generating a single extraction script applicable to all similar webpages within the group (pt. 2). Extraction results are then compared against human annotations for labeled data and synthetic annotations for unlabeled CommonCrawl webpages. The resulting scores are used to update the model weights (pt. 3). At inference time, SCRIBES enables the model to generalize to *new, unseen websites* by generating scripts that can be applied across similar webpages (pt. 4).

only negligible resource cost compared with running an LLM-based extraction on every individual page.

Although the idea appears straightforward, current LLMs struggle to produce high-quality, generalizable extraction scripts. Fine-tuning them for this ability is cumbersome, as creating annotations for such scripts is difficult even for expert labelers. The success of SCRIBES lies in a Reinforcement Learning (RL) framework that leverages structural similarities across related webpages: given a group of similar webpages, the model is rewarded when a script generated for one webpage also works on others. This encourages learning scripts that generalize beyond individual examples.

SCRIBES draws training data from two sources. First, it learns from *a small set of annotated examples* (192 pages from 34 groups) (Figure 1, parts 1–3). For each group, SCRIBES takes one webpage as input and prompts the model to generate a script intended to generalize across the group. The script is then executed on the remaining pages, and its outputs are compared with annotations to compute the reward. Second, SCRIBES leverages *in-the-wild websites from CommonCrawl* to further enhance its capabilities. We develop an iterative approach that starts from a checkpoint trained on annotated data and then refines the model to continue learning from their failed predictions on the in-the-wild websites. To provide supervision at scale, we employ LLM-based direct extractions as synthetic annotations, reducing reliance on annotations or hand-crafted parsers.

Extensive experiments show that our RL-trained model outperforms strong agentic baselines by more than 13% in generating robust, reusable parsing scripts. Moreover, we demonstrate that *improved extraction translates into downstream benefits*: in QA tasks requiring structured reasoning over HTML, incorporating triples produced by SCRIBES boosts accuracy across a wide range of LLMs, including SOTA models such as GPT-4o by over 4%.

## 2 RELATED WORKS

### 2.1 SEMI-STRUCTURED DATA PROCESSING

**Flattening**: In complex QA or retrieval settings that mix texts, tables, and knowledge bases, a common practice is to "linearize" everything into plain text (Oguz et al., 2022; Zhang et al., 2024;

Ma et al., 2022; Christmann et al., 2022). This is also a popular practice when dealing with HTML pages. Trafilatura is a widely used HTML cleaning and text extraction toolkit designed for large-scale web processing (Barbaresi, 2021), among many other HTML conversion packages (Firecrawl, 2025; Paraschiv, 2024). While effective for general text extraction, these utilities typically discard or flatten structural elements such as tables, lists, and infoboxes. Similar to findings in complex QA that highlight the importance of structural cues (Liu et al., 2024b; Zhang et al., 2024), recent work on RAG with raw HTML shows that converting to plain text discards headings, table structures, and other layout information critical for downstream tasks (Tan et al., 2025).

**Traditional IE Methods**: A classical approach to extracting structured data from semi-structured web content is wrapper induction, which learns extraction procedures ("wrappers") from a small set of labeled examples instead of hand-crafted rules (Kushmerick et al., 1997). Extensions include boosted wrapper induction, which combines simple patterns for greater robustness (Freitag & Kushmerick, 2000), and large-scale methods that handle noisy data and template drift (Dalvi et al., 2011). While effective on regular site structures with clean annotations, these methods are brittle to structural changes and generalize poorly across diverse domains. In contrast, our approach learns executable scripts, i.e. full extraction programs that operate directly on raw HTML, allowing the system to generalize beyond fixed rules and adapt automatically without manual template design.

**LLM-based methods:** Several recent advances utilize LLMs to extract semi-structured contents. For instance, Wang et al. (2025) train a LLM to convert HTMLs into Markdown and JSON using SFT and RL methods. Similarly, Poznanski et al. (2025) use a VLM to convert PDFs into clean, readable format retaining tabular structures. Many related works also exist on LLM-assisted knowledge-base construction (Gutiérrez et al., 2024; Zhang & Soh, 2024; Ning et al., 2023; Chen & Bertozzi, 2023; Zhang et al., 2023; Bai et al., 2025). However, calling an LLM per page remains resource-intensive at web-scale; moreover, they typically treat each page independently, missing the cross-page layout regularities that SCRIBES exploits.

## 2.2 RL WITHOUT ANNOTATIONS

A growing body of work explores reinforcement learning in settings without explicit annotations. Zuo et al. (2025) show that models can refine themselves at test time by turning consensus among rollouts into rewards, while Zhao et al. (2025) and Prabhudesai et al. (2025) demonstrate that internal signals such as self-certainty or confidence are sufficient to drive continued improvement. Shao et al. (2025) find that even spurious or random rewards can produce surprising gains, suggesting that models can bootstrap from imperfect signals. Like prior work, we reduce dependence on annotations by iteratively refining the model from its own failures, but instead of relying solely on internal signals, we utilize LLM-based direct extractions as synthetic annotation for reward calculation.

## 3 SCRIBES FRAMEWORK

### 3.1 PROBLEM DEFINITION

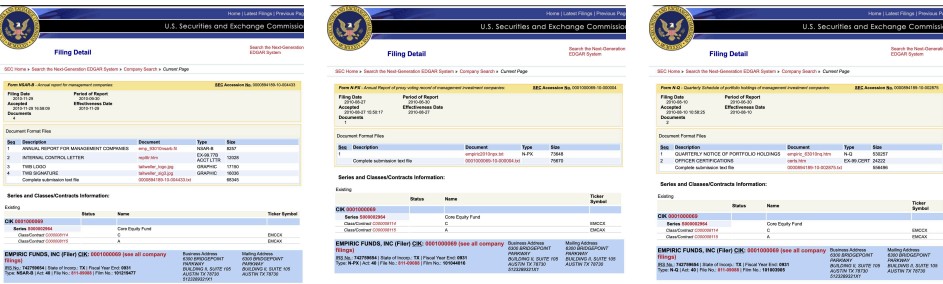

Figure 2: Three webpages containing semi-structured content under the same website.

**Knowledge extraction:** Let $G = \{p_1, \cdots, p_n\}$ be a group of semi-structured webpages that are structurally similar. The *knowledge extraction* task parses each page $p_i, i \in [1, n]$, to a list of triples (subjects, predicates, and objects). We denote by $y^\star_{p_i}$ the ground truth triples for page $p_i$.

**Extraction script generation:** We propose to solve the knowledge extraction problem by generating an extraction script that applies to every page in $G$. Formally, our goal is to train a model $LM$ that, given any webpage $p \in G$, predicts an extraction script $\hat{y}_p = LM(p)$, such that applying $\hat{y}$ to every page in $G$ generates triples close to ground truth triples $\{y_{p_i}^\star | p_i \in G\}$. For instance, in Figure 2, a model-generated script should robustly handle variations across webpages, such as differences in table sizes and values.

## 3.2 HTML DEDUPLICATION (DEDUP)

The raw HTMLs of webpages are typically very long and can easily surpass the maximum context window of even the long-context LLMs. We propose a simple yet effective method for deduplicating HTMLs: repeated HTML blocks are collapsed into a compact representation of the form "$n$ more …elements," which substantially reduces context length. Ablation experiments confirm that this deduplication step significantly improves model performance. We therefore apply it throughout our SCRIBES-trained models. An example of the dedup process is shown in Figure 6, and further details and analysis are provided in Appendix C.

## 3.3 RL SETUP

Annotating such extraction scripts for training is challenging even for expert human annotators. To address this, rather than relying on demonstrations, we propose adopting *Reinforcement Learning with Verifiable Rewards (RLVR)* for this task.

We define $r(p \to q) = S(\hat{y}_p(q), y_q^\star) \in [0, 1]$ as the score obtained when the script $\hat{y}_p$ is executed on a (possibly different) page $q$, where $S$ is a scoring function that measures similarity between predicted and annotated tuples. To compute this score, we follow prior works (Liu et al., 2024a; Sun et al., 2025) and adopt a bipartite matching algorithm that aligns predicted triples with gold triples by maximizing their pairwise fuzzy matching score. Based on this matching, we compute fuzzy precision $P^{\text{fuzzy}}$, recall $R^{\text{fuzzy}}$, and $F_1$ score $F_1^{\text{fuzzy}}$. Since fuzzy string similarity may fail to fully capture semantic equivalence, we additionally employ an LLM-as-a-judge (set to `Llama-3.3-70B-Instruct`) to evaluate the aligned triples (Prompt 17). We choose Llama to ensure consistency with prior work (Sun et al., 2025) and, by fixing the checkpoint, to enable reproducible experiments. This yields LLM-based precision $P^{\text{LM}}$, recall $R^{\text{LM}}$, and $F_1$ score $F_1^{\text{LM}}$. During training, we set $S = F_1^{\text{fuzzy}}$, the triple-level fuzzy $F_1$ score. Refer to Appendix F for additional details on metrics and an optimized implementation of $F_1^{\text{fuzzy}}$ during training.

### 3.3.1 REWARD SIGNAL FROM LABELED DATA

We define the following notations:

1. the *self-score* is $r_{\text{self}}(p) = r(p \to p)$, while
2. each *cross-score* is $r_{\text{cross}}(p, q) = r(p \to q)$ for $q \neq p$.

SCRIBES optimizes a model using Group Relative Policy Optimization (GRPO) (Shao et al., 2024) based on the following reward function for each training sample $p$:

$$r_{\text{SCRIBES}}(p) = \frac{1}{|G(p)|} \sum_{q \in G(p)} r(p \to q) = \frac{1}{|G(p)|} r_{\text{self}}(p) + \frac{|G(p)|-1}{|G(p)|} \sum_{q \in G(p), p \neq q} r_{\text{cross}}(p, q) \quad (1)$$

Within this framework, each self-score contributes only $\frac{1}{|G(p)|}$ to the final reward, while cross-scores constitute the majority of the reward signal. This design strongly encourages the model to generalize by accounting for potential variations across other, unseen webpages within the same group. We study the effect of different reward formulations through ablation studies in Section 4.4.

### 3.3.2 REWARD SIGNAL FROM UNLABELED DATA IN THE WILD

When training on annotated data, SCRIBES can directly leverage the gold human annotation $y_p^\star$ for each page $p$ as the reward signal. However, because the only high-quality annotated dataset available from Sun et al. (2025) is relatively small, it is inherently difficult to achieve broad coverage of diverse

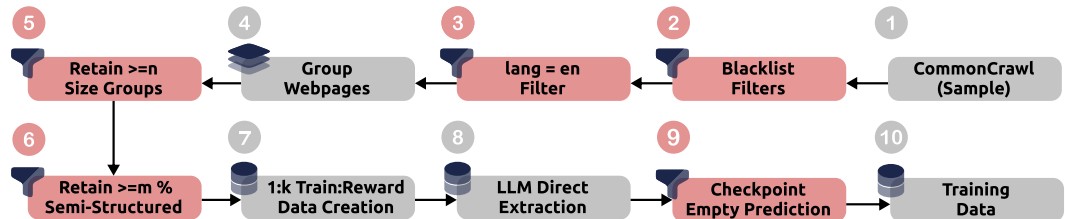

Figure 3: Processing pipeline for unlabeled data from CommonCrawl in Section 3.3.2.

website layouts using annotated data alone. To address this limitation, we propose a novel approach that leverages unlabeled in-the-wild webpages from CommonCrawl (abbreviated as CC) (Common Crawl, 2025).

Our data collection pipeline is illustrated in Figure 3. **(pt. 1)** Starting from a sample of CC, **(pt. 2)** we first apply the blacklist filters from Penedo et al. (2024) to remove adult or explicit content. **(pt. 3)** We then apply language filters to select English content websites and **(pt. 4)** group webpages by domain, **(pt. 5)** retaining only groups containing at least $n$ webpages. **(pt. 6)** Next, we use an LLM-based classifier (Prompt 15) to identify webpages containing semi-structured content, and we retain only those website groups where at least $m\%$ of the pages are classified as semi-structured. **(pt. 7)** Finally, we sample one webpage as the training example and associate it with up to $k \leq n$ in-group webpages for reward calculation. In our experiments, we apply the following thresholds: $n = 30$, $m = 90$, and $k = 13$.

At this stage, we obtain a collection of in-the-wild webpage groups containing semi-structured content. However, without human annotations, it is unclear what reward signal should be used for training. **(pt. 8)** To address this, we propose using LLM-based direct extraction (Prompt 16) as a proxy for gold annotations. Our experiments show this to be the strongest baseline. Nevertheless, because such direct extraction is far from perfect (achieving only about 40% $F_1$ for the best baseline), we aim to prevent noisy rewards from degrading model performance. **(pt. 9)** To this end, we start from a checkpoint trained on annotated data and identify a subset of webpages where the model's predicted scripts fail to produce any results. By concentrating training on these failure cases, we increase the likelihood that the additional synthetic data improves the model's performance. Ablation studies on the necessity of this subset are presented in Section 4.4.

# 4 EXPERIMENTS

## 4.1 DATASET

**Annotated dataset**: Existing datasets for semi-structured knowledge extraction from raw webpages are limited. *SemiBench* (Sun et al., 2025) presents a dataset of webpages drawn from 139 popular websites in CommonCrawl, annotated with triples. Their collection includes 83 websites with a single webpage, 46 groups of 3 similar webpages, and 10 groups of 13 similar webpages each. This grouping scheme provides a valuable opportunity to evaluate generalization in the SCRIBES setting. We select the 56 groups containing more than 1 webpage each for experiments in this work. We divided the annotated dataset into training and test sets using a 60%-40% split **across groups**; that is, we assign entire groups to either the training or test set, and we do not split within any group. For a group of size $n$ in the training/test set, we create $n$ training/test examples, each using one webpage as input and all group elements used for reward calculation. All evaluation metrics are reported on the test set, which contains only websites from groups that the model did not see during training. Refer to additional details in Appendix D.1.

**In-the-wild webpages**: To construct groups directly from CommonCrawl, we employ a simple heuristic: two webpages are grouped together if they share the same URL prefix up to the final substring. For example, `example.com/mid1/sub1` and `example.com/mid1/sub2` belong to the same group, while `example.com/mid2` does not. The LLM used in our pipeline is `GPT-OSS-120B`. We randomly sampled 50 webpages and estimated classifier accuracy at 90.0% precision and 72.0% recall. In total, 19,566 groups satisfied the $n \geq 30$ condition, among which 2,003 also satisfied the $m \geq 90$ condition. After direct extraction with the LLM, 1,898 examples

| Model and Method | All | | | Example | | | Holdout | | |
|---|---|---|---|---|---|---|---|---|---|
| | $R^{\text{LM}}$ | $P^{\text{LM}}$ | $F_1^{\text{LM}}$ | $R^{\text{LM}}$ | $P^{\text{LM}}$ | $F_1^{\text{LM}}$ | $R^{\text{LM}}$ | $P^{\text{LM}}$ | $F_1^{\text{LM}}$ |
| Baselines (Direct LLM Extraction) | | | | | | | | | |
| L-70B (Sun et al., 2025) [*] | 24.3 | 15.7 | 19.1 | - | - | - | - | - | - |
| Fine-tuned L-70B (Sun et al., 2025) [*] | 21.4 | 27.1 | 23.9 | - | - | - | - | - | - |
| GPT-4o (Sun et al., 2025) [*] | 35.1 | 23.8 | 28.3 | - | - | - | - | - | - |
| Q-14B flatten | 30.5 | 36.5 | 29.9 | - | - | - | - | - | - |
| Q-32B flatten | 28.7 | 37.4 | 29.9 | - | - | - | - | - | - |
| GO-20B 2-shot flatten | 33.2 | **47.1** | 34.9 | - | - | - | - | - | - |
| GO-120B 2-shot flatten | **42.3** | 46.3 | **40.4** | - | - | - | - | - | - |
| Baselines (Script-gen) | | | | | | | | | |
| Q-14B agentic-3-iter 2-shot | 8.6 | 11.1 | 8.0 | 13.2 | 18.0 | 12.6 | 6.3 | 7.8 | 5.7 |
| L-70B agentic-3-iter | 10.1 | 15.5 | 10.5 | 16.7 | 23.8 | 16.8 | 6.9 | 11.2 | 7.4 |
| Q-72B agentic-3-iter 2-shot | 16.4 | 19.4 | 15.0 | 24.1 | 28.6 | 21.8 | 13.3 | 15.8 | 12.4 |
| Q-32B agentic-3-iter 2-shot | 18.6 | 27.2 | 19.4 | 24.5 | 34.8 | 25.9 | 15.8 | 23.9 | 16.4 |
| GO-20B agentic-3-iter | 24.7 | 23.2 | 20.9 | 29.3 | 26.4 | 27.7 | 22.5 | 21.8 | 18.9 |
| GPT-4o agentic-3-iter 2-shot | 26.0 | 33.0 | 24.4 | 33.0 | 36.5 | 31.2 | 22.5 | 31.3 | 21.1 |
| GO-120B agentic-3-iter 2-shot | **33.9** | **41.0** | **34.3** | **35.8** | **42.3** | **36.6** | **33.0** | **40.5** | **33.3** |
| SCRIBES (Script-gen) | | | | | | | | | |
| Q-14B | 23.0 | 24.3 | 19.9 | 31.2 | 29.8 | 26.7 | 19.0 | 21.7 | 16.7 |
| Q-14B (+ CC) | 25.2 | 23.0 | 21.8 | 34.9 | 31.0 | 30.0 | 20.5 | 19.1 | 17.7 |
| Q-32B | 29.9 | 31.5 | 28.1 | 32.0 | 33.9 | 30.3 | 28.8 | 30.3 | 26.8 |
| Q-32B (+ CC) | **37.4** | **36.0** | **33.2** | **39.5** | **35.5** | **34.6** | **36.2** | **36.2** | **32.4** |

Table 1: LLM-judged metrics are reported separately for *All*, *Examples* (the webpage model used to generate the script), and *Holdout* (similar webpages where the same script was applied). Columns show macro-averaged $P^{\text{LM}}$, $R^{\text{LM}}$, and $F_1^{\text{LM}}$. For each model and block, we report only the strongest baseline here. The full baseline results, including LLM-based agentic baselines HippoRAG (Gutiérrez et al., 2024) and AutoSchemaKG (Bai et al., 2025), which exhibit lower scores, are provided in Table 11 in Appendix G.4. ([*]) Numbers reported by Sun et al. (2025) are on the full set.

were retained (the remainder corresponding to prediction failures or empty outputs). This entire process used less than 1% of the CC-MAIN-2025-30 crawl. We hypothesize that this pipeline can be scaled to larger portions of CommonCrawl for broader coverage; in this paper, we focus on establishing its feasibility.

## 4.2 TRAINING SETUP AND BASELINES

**Training** We train `Qwen2.5-Instruct` family models and perform minimal hyperparameter tuning to ensure stability during model training. Refer to Appendix D for additional details.

**Baselines** We experiment with both SOTA close-source and open-source models, including: `gpt-4o`, `Llama-3.3-70B-instruct` (abbreviated as L-70B), `Qwen2.5-Instruct` (abbreviated as Q-xB) family, and `gpt-oss` (abbreviated as GO-xB) family. We implement the following baselines for comparison (Prompt 19). By default, all baselines use Dedup as the SCRIBES-trained models. We explore multiple configurations to construct strong baseline models.

1. *agentic-n-iter*: After the model outputs a script given an example, if the script fails to produce output or produces empty output, we feed the execution feedback to the model and ask it to retry. Otherwise we use the output script as prediction. We repeat this ReAct-style (Yao et al., 2022) procedure up to $n$ times;

2. *n-shot*: We feed in $n$ HTMLs and their corresponding gold extraction results as in-context learning examples;

3. *flatten*: We directly flatten the HTML[3] and use it as model's input. Note that there is no generalizability requirement or dedup involved in this setup.

4. Recent, SOTA LLM-based KG construction pipelines, including HippoRAG (Gutiérrez et al., 2024) and AutoSchemaKG (Bai et al., 2025). See Section G.1 for details.

---

[3] `BeautifulSoup(html_content, "html.parser").get_text()`

### 4.3 RESULTS

**RQ1**: Does SCRIBES framework bring improvements to models in terms of their capability to extract semi-structured data?

For each example $p$ in our test set, models generate a script $\hat{y}_p = LM(p)$ and we apply it to all examples in $G(p)$. We derive a score

$$S(p) = \frac{1}{|G(p)|} \sum_{q \in G(p)} S(\hat{y}_p, y_q^\star) \tag{2}$$

where we set $S$ to be recall, precision, or $F_1$ score, as defined in Section 3.3. We refer to this aggregate score as "All." To further investigate the performance gap between the example provided to the model ("Example") and the other webpages to which the model-generated script is applied ("Holdout"), we decompose the score in Eq. 2 into two separate components:

$$S_{\text{example}}(p) = S(\hat{y}_p, y_p^\star) \qquad S_{\text{holdout}}(p) = \frac{1}{|G(p)| - 1} \sum_{q \in G(p),\, q \neq p} S(\hat{y}_q, y_q^\star)$$

In Table 1, we report the macro average of $R^{\text{LM}}, P^{\text{LM}}, F_1^{\text{LM}}$ by averaging individual $S(p)$ scores. SCRIBES-trained models drastically outperform strong agentic baselines. The best Q-14B and Q-32B models outperform the few-shot agentic base model performance by 13.8% in $F_1^{\text{LM}}$, and our best Q-32B model performs on-par with the few-shot agentic GO-120B model.

**RQ2**: Does using SCRIBES enable resource-efficient, web-scale extraction?

To demonstrate the SCRIBES-framework's applicability to web-scale semi-structured content extraction, we evaluate on a leftover subset of CommonCrawl data that was not used in model training. To keep the experiment tractable, we capped each group at 30 webpages and required at least 13 webpages per group, meaning this evaluation covers only a tiny fraction of the available data. On this small subset with 113,129 webpages, our model extracted 2,788,760 triples. Remarkably, only 4,661 required direct model predictions, while the vast majority were generated automatically through model-produced scripts.

On average, processing a webpage with deduplicated HTML requires 8,879 tokens, whereas using flattened HTML requires 2,399 tokens. Let $\rho = \frac{8879}{2399} \approx 3.7$ denote this relative per-page token ratio.

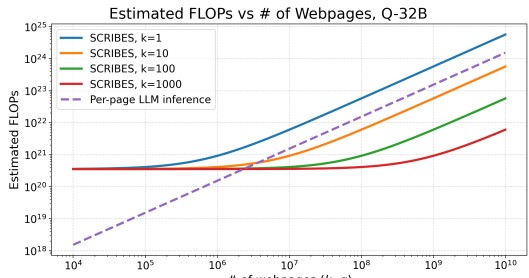

Figure 4: Estimated GPU FLOPs usage comparing SCRIBES-trained model with per-page LLM inference for Q-32B. The results show that, even when training compute is included, SCRIBES-trained models scale more efficiently at web scale.

Our approach quickly becomes more efficient as long as the target website contains at least 4 structurally similar pages. In fact, the token speedup of our scribe-based method relative to flattening grows linearly with $k$ (the number of structurally similar pages), following:

$$\text{speedup} = \frac{k}{\rho}$$

We further compare the total GPU cost of SCRIBES, including training, with per-page LLM inference in Figure 4. Let $g$ denote the number of groups processed. While per-page inference (dashed purple line) increases linearly with both the number of groups $g$ and the group size $k$, the SCRIBES-trained model yields substantial FLOP savings, with the magnitude of savings growing proportionally to group size. For instance, with 100 pages per group, SCRIBES can already provide a computational saving of $1.12 \times 10^{21}$ FLOPS when processing $10^5$ groups. Additional details on the FLOP estimates are provided in Appendix D.3

Thus, compared to approaches that require per-page LLM inference (Bai et al., 2025), SCRIBES can significantly cut down the GPU resource usage for web-scale extraction.

| Model and Method | All | | | Example | | | Holdout | | |
|---|---|---|---|---|---|---|---|---|---|
| | $R^{\mathrm{LM}}$ | $P^{\mathrm{LM}}$ | $F_1^{\mathrm{LM}}$ | $R^{\mathrm{LM}}$ | $P^{\mathrm{LM}}$ | $F_1^{\mathrm{LM}}$ | $R^{\mathrm{LM}}$ | $P^{\mathrm{LM}}$ | $F_1^{\mathrm{LM}}$ |
| Q-14B (Reward w/ Eq. 3) | 15.6 | 19.6 | 15.7 | 29.1 | **36.2** | **27.9** | 8.8 | 11.0 | 9.5 |
| Q-14B (SCRIBES) | **23.0** | **24.3** | **19.9** | **31.2** | 29.8 | 26.7 | **19.0** | **21.7** | **16.7** |

Table 2: Ablation study of reward design (Eq. 3), showing that SCRIBES 's reward significantly enhances performance on holdout webpages.

| Method | All | | | Example | | | Holdout | | |
|---|---|---|---|---|---|---|---|---|---|
| | $R^{\mathrm{LM}}$ | $P^{\mathrm{LM}}$ | $F_1^{\mathrm{LM}}$ | $R^{\mathrm{LM}}$ | $P^{\mathrm{LM}}$ | $F_1^{\mathrm{LM}}$ | $R^{\mathrm{LM}}$ | $P^{\mathrm{LM}}$ | $F_1^{\mathrm{LM}}$ |
| Q-14B (Annotated only) | 23.0 | 24.3 | 19.9 | 31.2 | 29.8 | 26.7 | 19.0 | 21.7 | 16.7 |
| Q-14B (+ All CC) | 22.0 | **30.2** | **22.0** | 28.9 | 35.1 | 28.1 | 18.4 | **27.6** | **18.8** |
| Q-14B (+ Failure-Case CC) | **25.2** | 23.0 | 21.8 | **34.9** | 31.0 | **30.0** | **20.5** | 19.1 | 17.7 |
| Q-32B (Annotated only) | 29.9 | 31.5 | 28.1 | 32.0 | 33.9 | 30.3 | 28.8 | 30.3 | 26.8 |
| Q-32B (+ All CC) | 31.1 | 34.1 | 29.7 | 35.2 | **37.0** | 36.1 | 32.9 | 29.0 | 28.1 |
| Q-32B (+ Failure-Case CC) | **37.4** | **36.0** | **33.2** | **39.5** | 35.5 | **34.6** | **36.2** | **36.2** | **32.4** |

Table 3: Ablation study on CC data subsets, showing that models trained with the failure-case subset generally perform better.

| Model and Method | All | | | Example | | | Holdout | | |
|---|---|---|---|---|---|---|---|---|---|
| | $R^{\mathrm{LM}}$ | $P^{\mathrm{LM}}$ | $F_1^{\mathrm{LM}}$ | $R^{\mathrm{LM}}$ | $P^{\mathrm{LM}}$ | $F_1^{\mathrm{LM}}$ | $R^{\mathrm{LM}}$ | $P^{\mathrm{LM}}$ | $F_1^{\mathrm{LM}}$ |
| Q-14B agentic 3-iter 2-shot | 9.5 | 13.7 | 8.8 | 23.2 | 24.2 | 20.0 | 12.4 | 7.4 | 7.2 |
| Q-14B (SCRIBES) | **20.7** | **22.2** | **19.4** | **31.8** | **36.1** | **30.4** | **14.5** | **12.0** | **12.2** |

Table 4: Ablation study on cross-domain transferability, showing that the SCRIBES-trained model demonstrate strong cross-domain transfer skills and outperform the baseline by more than 10%.

## 4.4 ABLATIONS

**RQ3**: Does the SCRIBES reward design improve the model's capability in generating scripts that generalize to holdout elements?

To answer this question, we train a Q-14B model with the following reward for each training example $p$:

$$r_0(p) = r_{\mathrm{self}}(p) \tag{3}$$

Compared to Equation 1, this reward encourages the model only to generate scripts suited to the current training example, without considering other in-group elements. We still use the same input prompt as in our SCRIBES-trained models (Prompt 19), which instructs the model to produce scripts that generalize across similar webpages. The training setup remains unchanged.

As shown in Table 2, although this model outperforms Q-14B (SCRIBES) on the examples encountered during inference ($+1.2\%$), it generalizes much more poorly to similar webpages where the script is applied ($-7.2\%$), resulting in worse overall performance in the "All" column ($-4.2\%$). This shows that the SCRIBES reward design can more effectively instill in models the capability to produce generalizable scripts.

**RQ4**: Does using CommonCrawl data bring further improvements to our models?

We apply the technique described in Section 3.3.2 to the final checkpoints of the SCRIBES-trained Q-14B and Q-32B models on the annotated dataset. As shown in Table 1, additional training on synthetic data derived from CommonCrawl further improves performance, yielding gains of roughly 2% for Q-14B and 5% for Q-32B overall.

To better understand the impact of noisy rewards, we conducted the following ablation studies: (1) training directly on CC data, and (2) training on a mixture of CC and annotated data at a 1:1 ratio. Neither approach led to performance improvements, as shown in Table 10 (Appendix G.2). We therefore hypothesize that it is essential to first train the model with gold rewards to establish strong prior knowledge of this task. Subsequent training with noisy rewards can then expose the model to

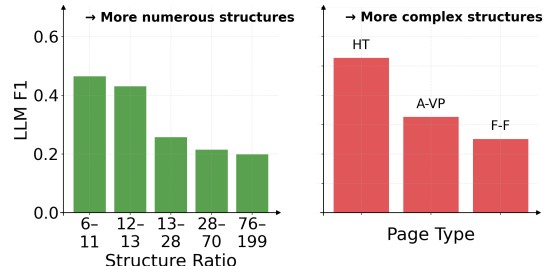

(a) Performance of our best Q-32B model by amount of structure and page type.

| Breakdown | $P^{LM}$ | $R^{LM}$ | $F_1^{LM}$ |
|---|---|---|---|
| Nested list | 23.6 | 18.7 | 19.7 |
| Multi-column | 23.3 | 33.9 | 18.6 |
| All | **39.5** | **35.5** | **34.6** |

(b) Comparison of the best Q-32 model's performance across nested-list and multi-column cases on our test set.
A nested list is defined as content containing a table or another list embedded within an outer list, while multi-column content refers to instances where headers span multiple columns.

Figure 5: Error analysis: (a) model performance by structure complexity; (b) comparison across nested lists and multi-column formats.

| Additional reference | Q-1.5B | Q-3B | Q-7B | Q-14B | Q-32B | GPT-4o |
|---|---|---|---|---|---|---|
| Flattened HTML | 50.2 | 53.8 | 62.9 | 74.2 | 70.8 | 82.5 |
| + Best Direct LLM Extraction triples | 52.8 | 61.2 | 66.6 | 73.5 | 73.1 | 82.7 |
| + Best Q-32B triples | 52.9 | 54.3 | 64.1 | 77.3 | 73.2 | 86.6 |
| + Ground truth triples | 60.5 | 64.9 | 70.5 | 78.2 | 74.8 | 87.4 |

Table 5: QA accuracy (%) with triple augmentations (evaluated by `Llama-3.3-instruct-70B`, Prompt 20). SCRIBES 's predicted triples boost QA performance across many models.

more diverse inputs, not only preserving but further improving performance, analogous to findings in Shao et al. (2025).

**RQ5**: What's the effect of selecting the failure case subset to continue CommonCrawl trainings?

As discussed in Section 3.3.2, we select the subset of CC data where our model produced scripts with no valid triples extracted. We examine whether restricting training to this subset is necessary by training both a 14B and a 32B model on the full CC dataset ("All CC") and only the subset where no triples were extracted ("Failure-Case CC"). Results are reported in Table 3. We highlight two findings: (1) Training on either All CC or Failure-Case CC improves performance compared to using annotated data alone, and (2) Failure-Case CC yields stronger gains for Q-32B compared to All CC (+3.5%) , while performance for Q-14B remains comparable across the two settings.

**RQ6**: Do SCRIBES-trained models transfer across domains? For example, does a model trained on finance or legal tables generalize to product or encyclopedia pages?

To investigate this question, we conduct an ablation study using a train–test split in which the test set contains all product and encyclopedia pages, while the training set excludes webpages from these domains entirely. Details on this setup are provided in Appendix G.3. As shown in Table 4, the SCRIBES-trained model still substantially outperforms the strongest agentic baseline of the same model by more than 10%. To develop a model capable of web-scale extraction, we would still recommend training on a dataset that encompasses diverse domains and page layouts, as demonstrated by our CommonCrawl processing in Section 3.3.2.

## 4.5 ERROR ANALYSIS

We perform an error analysis to understand the failures of the best-performing Q-32B model. We break down performance by the amount of structure in a webpage (approximated by the ratio of raw HTML length to flattened text length) and by webpage type. As shown on the left of Figure 5a where webpages are grouped into five equal-sized bins (by number of webpages) and the respective medians are reported, performance declines as webpages contain more structure. On the right, the model performs best on webpages with Horizontal Tables (HT), followed by Attribute–Value Pairs (A-VP), and performs worst on Free-Form (F-F) pages. These results suggest that webpages with more numerous or complex structures are particularly challenging for our model.

We also compare the performance of our model's outputs on contents involving multi-column and nested lists. As shown in Table 5b, we observe that such content is more challenging for our model. Further prediction examples are showcased in Appendix I.

# 5 Downstream Applications

## 5.1 Question Answering over Semi-Structured Web Data

We demonstrate that our script-extracted triples can enhance QA performance, even for the most capable LLMs. Although there exist many general-purpose QA datasets (Yang et al., 2018; Rajpurkar et al., 2016) and datasets focused on semi-structured databases (Chen et al., 2020; Zhu et al., 2021; Chen et al., 2021), very few address the setting where the input consists of raw HTML. SemiBench (Sun et al., 2025) fills this gap, containing QA pairs with aligned triple annotations. This makes it a strong testbed for evaluating whether triple extraction improves QA over semi-structured web data. We select the subset of QA data (a total of 416 QA pairs) associated with our test set and evaluate a broad range of models as QA backbones, using the following reference conditions in Prompt 18: (1) Flattened HTML only; (2) Flattened HTML with best-performing direct LLM-extracted triples (GO-120B 2-shot flatten); (3) Flattened HTML with our model-extracted triples; and (4) Flattened HTML with gold triples. We report the result on the QA pairs associated with our validation examples in Table 5. Our SCRIBES-trained models yield consistent gains across diverse QA backbones, including an improvement of more than 4% for GPT-4o.

We further observe that although the SCRIBES-trained models slightly underperform the strongest per-page LLM-inference baseline in Table 1, they nonetheless deliver comparable downstream QA gains. As shown in Table 5, using SCRIBES-generated triples improves QA performance for Q-14B and GPT-4o, yields roughly similar performance for Q-1.5B and Q-32B, and performs worse for Q-3B and Q-7B. These results indicate that higher IE accuracy does not necessarily translate into better downstream QA performance. Instead, using SCRIBES-produced triples can deliver much better efficiency and a similar level of downstream QA improvement.

## 5.2 Further Discussions

The efficiency benefits of SCRIBES open up additional opportunities, and we highlight two directions for future explorations:

**Multi-page, Complex QAs**: SCRIBES-extracted triples enable queries that require aggregation or ranking across multiple webpages. For example, a standard RAG solution would struggle with questions like "What is the latest report filed?" when answering against the website in Figure 2. In contrast, SCRIBES-generated triples can efficiently support such queries, eliminating the need for resource-intensive, page-by-page KG construction with LLMs.

**Pretraining**: Most open-source pretraining corpora systematically filter out semi-structured content. For instance, C4 (Raffel et al., 2023) applies a "punctuation filter" that removes sentences not ending with valid punctuation. Recent popular corpora such as Dolma (Soldaini et al., 2024) and FineWeb (Penedo et al., 2024) inherit this bias, resulting in a near-complete absence of semi-structured data. We believe SCRIBES can address this gap by enabling efficient and resource-effective extraction and incorporation of such content into pretraining datasets.

# 6 Conclusion

This work introduces a novel RL framework, SCRIBES, for training models to generate generalizable extraction scripts across structurally similar webpages for semi-structured content extraction. We also propose a new method for generating synthetic training data, which further improves model performance, by leveraging in-the-wild webpages from CommonCrawl. Experiments on our dataset demonstrate that SCRIBES-trained models yield substantial gains in question answering over semi-structured data. We hope that SCRIBES will facilitate further research on semi-structured content, such as complex QA and pretraining, and serve as a valuable tool for the community.

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

APPENDIX

## A  USE OF LLMs IN THIS RESEARCH

We utilize LLMs in two main ways in this research:

1. **Assistance with Code Writing:** During the implementation of RL training and evaluation scripts, LLMs were occasionally used as assistants. All code was subsequently double-checked and verified by the authors.

2. **Paper Language and Related Works:** During the writing process, we occasionally utilized LLMs to improve the clarity and fluency of the English. We also occasionally use LLM-assisted search systems to find additional related works. All final text was reviewed by the authors.

## B  WEBSITES WITH SEMI-STRUCTURED CONTENT

We can broadly classify webpages with semi-structured content into three categories:

1. **Horizontal Tables**: These webpages primarily present information in a tabular format.

2. **Attribute-Value Pairs**: Information is organized as attribute-value pairs, typically displayed across multiple rows in an "infobox"-like format.

3. **Free Form**: Semi-structured content is distributed throughout the page, often combining both horizontal tables and attribute-value pairs.

For additional information and more details on these breakdowns, refer to Sun et al. (2025).

## C  HTML DEDUP ALGORITHM DETAILS

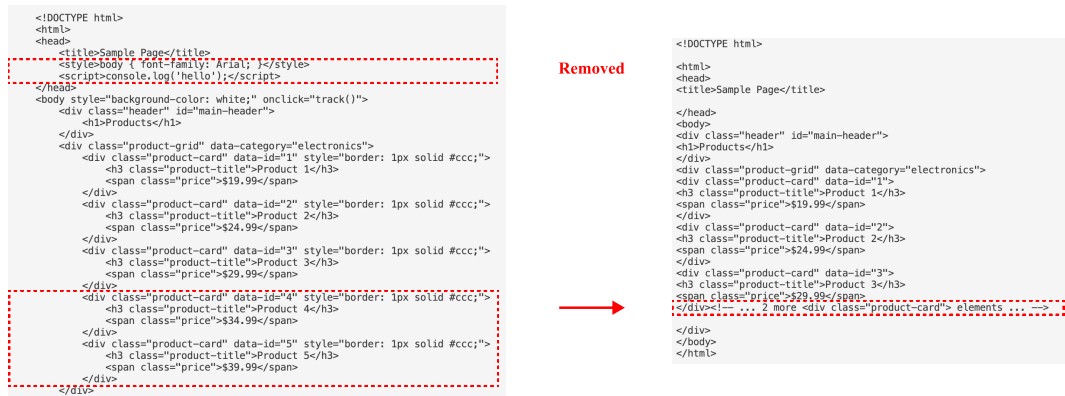

Figure 6: An example illustrating Algorithm 1 is shown here. The original HTML appears on the left, while the compressed HTML is shown on the right. The dashed-highlighted section near the top, containing script and style elements, has been removed. The repeated HTML content near the bottom has been deduplicated, retaining up to $z = 3$ elements.

Raw HTMLs are often long and repetitive. We propose a simple and effective dedup algorithm to significantly cut down the token length of HTML pages while still maintaining its structure. Algorithm 1 shows the implementation of this algorithm. We set $z = 3$ in our experiments.

Table 6 shows the token saving effect of our dedup algorithm. Removing whitespaces in a HTML only brings minimal token savings ($< 2\%$), while our dedup algorithm brings significant token savings, cutting down token usage from $>114$k to $<17$k. We also profiled performance gains of baselines models using dedup. As shown in Table 7, employing deduplicated HTML yields clear improvements compared to using raw HTML. Most notably, deduplication significantly increases the Non-Empty Rate of baseline performance by enabling more data points to fit within the model's context window.

---

**Algorithm 1** Structure-Preserving HTML Deduplication (keep-$z$)

---

**Require:** Raw HTML string $H$, integer $z \geq 1$ (default $z$=3)
**Ensure:** Compressed, structure-preserving HTML
 1: Parse $H$ into DOM $R$ (fallback parser if needed; return $H$ on failure)
 2: RemoveTags ← {script, style, noscript,
        iframe, embed, object, applet,
        meta, link, base}
 3: KeepAttrs ← {id, class, role, name,
        type, href, src, alt, title,
        rel, target, for, action, method,
        value, placeholder, required, data-*, aria-*}
 4: Remove all nodes with tag in RemoveTags
 5: Remove all HTML comments except those starting with "..."
 6: **for all** element nodes $e$ in $R$ **do**
 7:     **for all** attributes $a$ of $e$ **do**
 8:         **if** $a \notin$ KeepAttrs and $a$ not prefixed by data- or aria- **then**
 9:             delete attribute $a$ from $e$
10:         **end if**
11:     **end for**
12: **end for**
13: **for all** nodes $n$ in traversal of $R$ **do**
14:     **if** $n$.tag ∈ {ul, ol, div, section, tbody, thead, select} **then**
15:         children ← [ $c \in n$.children : $c$ is an element ]
16:         Group children by sig($c$) ← ($c$.tag, sort($c$.class or [ ]))
17:         **for all** group $G$ **do**
18:             **if** $|G| > z$ **then**
19:                 Keep the first $z$ in $G$ (order preserved); remove the rest
20:                 After the $z$-th kept node, insert comment:
21:                     " ... $|G| - z$ more <tag class='...'> elements ... "
22:             **end if**
23:         **end for**
24:     **end if**
25: **end for**
26: Optionally normalize whitespace and excessive blank lines
27: **return** serialized DOM

---

| Processing Stage | Avg Tokens | Percentage |
|---|---|---|
| Original tokens | 114,318.6 | 100.0% |
| After whitespace removal | 112,279.0 | 98.2% |
| After dedup | 16,985.1 | 14.9% |
| **Reductions** | | |
| Whitespace token savings | 2,039.6 | 1.8% |
| Total dedup token savings | 97,333.5 | 85.1% |

Table 6: Token reduction analysis across the webpages collected by Sun et al. (2025). Tokens were profiled with GPT-4o tokenizer, accessed via https://github.com/openai/tiktoken.

## D   TRAINING HYPERPARAMETERS AND OTHER DETAILS

### D.1   DATA PRE-PROCESSING

During training, we set the maximum prompt length to 28672 tokens and the maximum response length to 4096 tokens. This results in a total model context window of 32768 tokens, which is the maximum length before needing to apply YaRN (Peng et al., 2023) for the Qwen-2.5 series models[4].

---

[4]We observed empirically that model training with YaRN becomes much more unstable and difficult to converge.

| Model & Format | $P^{\text{LM}}$ | $R^{\text{LM}}$ | $F_1^{\text{H, LM}}$ | Non-Empty Rate |
|---|---|---|---|---|
| L-70B w/ Raw HTML | 3.4 | 3.7 | 3.5 | 37.9 |
| L-70B w/ Dedup HTML | 14.2 | 9.5 | 11.3 | 46.4 |
| GPT-4o w/ Raw HTML | 13.7 | 15.4 | 14.5 | 63.8 |
| GPT-4o w/ Dedup HTML | 19.1 | 23.0 | 20.9 | 94.9 |

Table 7: Performance comparison of baseline models using raw or dedup-ed HTML. Here, we feed each page in one-by-one in this dataset and only evaluate the model's performance on one given page. Non-Empty Rate is set to 1 if the model's generated code produced at least 1 triple on this page, and 0 if otherwise.

SemiBench (Sun et al., 2025) includes a subset of 268 webpages drawn from 56 groups, each containing more than one webpage. We partition the groups into training and test sets at an approximately 6:4 ratio, resulting in 34 groups (192 webpages) for training and 22 groups (76 webpages) for testing. After applying the maximum-context constraint described above, 141 training webpages and 65 test webpages remain.

## D.2 TRAINING DETAILS

During GRPO training, we do not apply entropy loss. We set the KL loss coefficient to 0.001 and the KL loss to be the $k_3$ loss using the approximation described in Schulman (2020), i.e.,

$$k_3(a) = \frac{\pi_{\text{new}}(a)}{\pi_{\text{old}}(a)} - \log \frac{\pi_{\text{new}}(a)}{\pi_{\text{old}}(a)} - 1$$

We use the default model rollout parameters (for Qwen-2.5-instruct, these are top_k= $-1$, top_p= 1, and temperature = 1) and validation/inference parameters (for Qwen-2.5-instruct, these are top_k= $-1$, top_p= 1, and temperature = 0). We do not use LoRA and instead perform full-parameter finetuning with FSDP (Zhao et al., 2023). We trained the models on the annotated set for a total of 50 epochs, and on CommonCrawl data for 1 epoch. For each update, we collect 8 rollouts to perform GRPO update. For the 32B model, we apply a 0.5 gradient clipping, which we found to lead to more stable trainings. We set the learning rate to be a constant $1e - 6$.

## D.3 DETAILED ON COMPUTE COMPARISON

To train the Q-14B SCRIBES model, we ran approximately 12 hours of training at an average throughput of 3,958 TFLOPs across all GPUs, yielding an estimated total training compute of $1.71 \times 10^{20}$ FLOPs. For the Q-32B SCRIBES model, we similarly trained for about 12 hours at an average of 8,232 TFLOPs, resulting in an estimated total compute of $3.56 \times 10^{20}$ FLOPs.

To estimate the per-page inference cost reported in Figure 4, we assume that each forward pass requires roughly 2 times the parameter count per token.

## E DATASET COMPARISON

We compare several statistics of HTML webpages in Table 8. Below, we define each statistic:

**DOM Max Depth**: The maximum depth of the Document Object Model (DOM) tree in an HTML document. This measures how deeply the elements are nested; a higher DOM Max Depth indicates more extensive nesting.

**Deduplication Ratio**: The lengths of the HTML content before and after applying the deduplication algorithm described in Appendix C (in characters). This quantifies redundancy in the HTML structure; a lower Deduplication Ratio indicates greater redundancy.

**Structure Ratio**: The ratio of the HTML length to the flattened text length (in characters). This approximates how much structural markup the HTML contains relative to its textual content; a higher Structure Ratio reflects more structural complexity.

**Tag Count**: The number of all tags in an HTML document[5]. This measures the structural complexity of the HTML; a higher Tag Count indicates a more complex document.

| Feature | Metric | Train | Test | CC (After Step 6 in Fig. 3) |
|---|---|---|---|---|
| DOM Max Depth | Mean | 20.2 | 18.4 | 20.2 |
| | Median | 19.0 | 17.0 | 17.0 |
| | Std | 4.94 | 6.98 | 21.7 |
| | Min | 10.0 | 10.0 | 5.00 |
| | Max | 37.0 | 37.0 | 455 |
| Deduplication Ratio | Mean | 0.215 | 0.174 | 0.353 |
| | Median | 0.213 | 0.166 | 0.344 |
| | Std | 0.111 | 0.0986 | 0.178 |
| | Min | 0.0302 | 0.0324 | 0.000480 |
| | Max | 0.484 | 0.422 | 1.02 |
| Structure Ratio | Mean | 46.1 | 43.5 | 20.8 |
| | Median | 28.7 | 27.4 | 13.8 |
| | Std | 40.0 | 45.5 | 48.7 |
| | Min | 2.24 | 6.00 | 1.19 |
| | Max | 174 | 199 | 1960 |
| Tag Count | Mean | 1650 | 1820 | 655 |
| | Median | 1260 | 1080 | 496 |
| | Std | 2140 | 2550 | 559 |
| | Min | 224 | 154 | 18.0 |
| | Max | 27800 | 12300 | 5070 |

Table 8: Summary statistics (Mean, Median, Std, Min, Max) for HTML-derived features across datasets.

This comparison shows that the labeled training and test sets share similar summary statistics, whereas the CommonCrawl portion differs noticeably. In particular, the CommonCrawl data is less redundant (lower Deduplication Ratio), contains less structural markup (lower Structure Ratio), and is structurally simpler (lower Tag Count). Across all metrics, it also exhibits greater variability, as indicated by the higher standard deviations. These observations suggest that incorporating this portion of the CommonCrawl data into training can meaningfully broaden the distribution of inputs, exposing the models to examples that differ substantially from those in the labeled dataset.

# F    METRICS AND THEIR IMPLEMENTATION

## F.1    DETAILS ON THE FUZZY MATCH ALGORITHM

Formally, let $G = \{g_1, g_2, \ldots, g_m\}$ denote the set of gold triples and $P = \{p_1, p_2, \ldots, p_n\}$ the predicted triples. Instead of requiring exact equality, we define a similarity function $f^{\text{fuzzy}}(g_i, p_j) \in [0, 1]$ that quantifies the degree of match between a gold triple $g_i$ and a predicted triple $p_j$ as the ratio of character-level matching[6]. To ensure one-to-one alignment, we compute a maximum-weight bipartite matching between $G$ and $P$, where the weight of each edge is $f^{\text{fuzzy}}(g_i, p_j)$. This assignment is efficiently solved using the Jonker–Volgenant algorithm[7]. Precision, recall, and $F_1$ are then generalized as:

$$P^{\text{fuzzy}} = \frac{\sum_{(g,p)\in M} f^{\text{fuzzy}}(g,p)}{|P|}, \quad R^{\text{fuzzy}} = \frac{\sum_{(g,p)\in M} f^{\text{fuzzy}}(g,p)}{|G|}, \quad F_1^{\text{fuzzy}} = \frac{2 \cdot P^{\text{fuzzy}} \cdot R^{\text{fuzzy}}}{P^{\text{fuzzy}} + R^{\text{fuzzy}}}.$$

---

[5] `len(soup.find_all(True))`

[6] Implemented via `https://github.com/seatgeek/fuzzywuzzy`'s ratio function, which calculate a ratio of character-level matching using Levenshtein distance .

[7] Implemented via `https://docs.scipy.org/doc/scipy/reference/generated/scipy.optimize.linear_sum_assignment.html`.

| Method | $R^{\text{LM}}$ | $P^{\text{LM}}$ | $F_1^{\text{LM}}$ | $R^{\text{fuzzy}}$ | $P^{\text{fuzzy}}$ | $F_1^{\text{fuzzy}}$ |
|---|---|---|---|---|---|---|
| Q-14B flatten | 30.46 | 36.46 | 29.87 | 45.96 | 52.37 | 43.50 |
| Q-32B flatten | 28.73 | 37.44 | 29.93 | 41.62 | 54.25 | 42.26 |
| GO-20B 2-shot flatten | 33.18 | 47.10 | 34.93 | 46.53 | 65.21 | 49.77 |
| GO-120B 2-shot flatten | 42.27 | 46.26 | 40.40 | 56.01 | 61.42 | 53.37 |
| Q-14B 3-iter 2-shot | 8.59 | 11.13 | 8.01 | 17.17 | 25.57 | 16.53 |
| Q-72B 3-iter 2-shot | 16.40 | 19.41 | 14.97 | 28.73 | 37.96 | 28.60 |
| Q-32B 3-iter 2-shot | 18.56 | 27.20 | 19.41 | 27.49 | 44.67 | 30.39 |
| GO-20B 3-iter | 24.70 | 23.22 | 20.87 | 52.30 | 41.83 | 39.58 |
| GPT-4o 3-iter 2-shot | 25.95 | 33.04 | 24.42 | 45.58 | 60.57 | 44.46 |
| GO-120B 3-iter 2-shot | 33.86 | 40.96 | 34.30 | 49.79 | 65.72 | 52.02 |

Table 9: Comparison of LLM-judged metrics and fuzzy-matching metrics for baselines reported in Table 1 for the "All" column. Gray-highlighted columns denote $F_1^{\text{LM}}$ and $F_1^{\text{fuzzy}}$. This comparison shows that the two metrics show similar performance trend across models and configurations.

where $M \subseteq G \times P$ denotes the optimal matching. Given $M$, the LLM-based metric evaluates correctness by invoking a LLM on the final matched pairs of gold and predicted triples. For each pair $(g, p) \in M$, the model outputs a binary judgment $f^{\text{LM}}(g, p) \in \{0, 1\}$, where 1 denotes a true match and 0 denotes a failed match according to Prompt 17. We then define LLM-based precision, recall, and $F_1$ as:

$$P^{\text{LM}} = \frac{\sum_{(g,p) \in M} f^{\text{LM}}(g, p)}{|P|}, \quad R^{\text{LM}} = \frac{\sum_{(g,p) \in M} f^{\text{LM}}(g, p)}{|G|}, \quad F_1^{\text{LM}} = \frac{2 \cdot P^{\text{LM}} \cdot R^{\text{LM}}}{P^{\text{LM}} + R^{\text{LM}}}.$$

Empirically, we observe a correlation between $F_1^{\text{fuzzy}}$ and $F_1^{\text{LM}}$. The latter tends to yield slightly lower absolute scores but exhibits the same performance trend across models and configurations. A comparison showing the two metrics and the associated precision and recall metrics for the baselines are shown in Table 9. We calculated the correlation coefficient between $F_1^{\text{fuzzy}}$ and $F_1^{\text{LM}}$ to be $0.957$ with a p-value of $1.4 \times 10^{-5}$, showing a strong positive correlation.

### F.2 REWARD DURING RL IMPLEMENTATION

We use $F_1^{\text{fuzzy}}$ during training as a proxy for $F_1^{\text{LM}}$, thereby avoiding LLM calls. Because computing fuzzy $F_1$ exactly requires solving a maximum-weight bipartite matching, runtime can become too long for large sets of triples. We thus approximate the matching with a greedy heuristic. Specifically, all candidate pairs of gold and predicted triples are scored by $f^{\text{fuzzy}}$, sorted in descending order, and added sequentially to the matching as long as they do not conflict with previously chosen pairs. This yields a fast, albeit sub-optimal, alignment. To ensure scalability, we impose a 60-seconds cutoff for evaluation. If timeout occurs, we further project the total similarity score by extrapolating from the average score of observed matches to the remaining unmatched capacity.

### F.3 HUMAN VERIFICATION OF RL REWARD

We followed the same evaluation metrics as defined in Sun et al. (2025), which reported a 95% agreement rate between the LLM-based F1 metric $F_1^{\text{LM}}$ and human judgments, indicating strong alignment.

## G ADDITIONAL EXPERIMENTS

### G.1 ADDITIONAL LLM-BASED AGENTIC BASELINES

In addition to the simple 2-shot baseline, we profile two promising LLM-based agentic knowledge-base–construction baselines: HippoRAG (Gutiérrez et al., 2024) and AutoSchemaKG (Bai et al., 2025), representative of recent LLM-driven KG construction pipelines.

HippoRAG is a retrieval-augmented generation framework that builds a knowledge graph as an embedding index, mimicking the role of the hippocampus in human memory. We use the first stage

| Method | All | | | Example | | | Holdout | | |
|---|---|---|---|---|---|---|---|---|---|
| | $R^{\text{LM}}$ | $P^{\text{LM}}$ | $F_1^{\text{LM}}$ | $R^{\text{LM}}$ | $P^{\text{LM}}$ | $F_1^{\text{LM}}$ | $R^{\text{LM}}$ | $P^{\text{LM}}$ | $F_1^{\text{LM}}$ |
| Q-14B (Annotated mixed with CC) | 6.5 | 8.0 | 6.5 | 8.1 | 9.6 | 7.9 | 5.7 | 6.4 | 5.7 |
| Q-14B (CC only) | 7.7 | 15.8 | 9.2 | 8.9 | 18.4 | 10.8 | 7.2 | 14.7 | 8.4 |
| Q-14B (Annotated followed by CC) | **25.2** | **23.0** | **21.8** | **34.9** | **31.0** | **30.0** | **20.5** | **19.1** | **17.7** |

Table 10: Ablation study on the impact of noisy reward. We compare three training configurations: (1) CC data only, (2) annotated data mixed with CC data at a 1:1 ratio, and (3) training first on annotated data followed by CC data. Results show that noisy reward alone or mixed training does not improve performance, whereas a staged setup, first training on annotated data before continuing with CC, yields substantial gains.

of their KG construction pipeline, which consists of two prompts, one for named entity recognition (NER) and one for triple extraction. We also replace their 1-shot example with the same 2-shot examples used in our baseline.

AutoSchemaKG is a framework for web-scale KG construction over a pretraining-scale corpus. It calls three LLM modules on each webpage: (1) an entity–entity relationship extractor, (2) an entity–event relationship extractor, and (3) an event–event relationship extractor. These prompts are all zero-shot and are challenging to adapt, so we retain them as originally specified.

As shown in Table 11, the simple 2-shot baseline outperforms both LLM-based baselines across all models evaluated on our task, including by more than 20% for the strongest model, GPT-OSS-120B. Moreover, they inherit the same cost inefficiencies, as each webpage requires multiple LLM calls.

### G.2 ADDITIONAL ABLATION EXPERIMENT ON IMPACT OF NOISY REWARD

To further investigate the role of noisy reward, we conduct additional ablation experiments under three training configurations: (1) training on CC data only, (2) training on a mixture of CC and annotated data at a 1:1 ratio, and (3) training first on annotated data and then continuing on CC data. Results are reported in Table 10.

### G.3 ADDITIONAL ABLATION EXPERIMENT ON DOMAIN TRANSFERABILITY

In this ablation study, we reorganized the dataset by assigning each website to one of the following content categories:

- Finance & Economics
- Legal & Regulatory
- Developer & Software
- Science & Research
- Science & Database
- Sports
- Gaming & Entertainment
- Media & Entertainment
- Real Estate
- Social Platforms
- Weather & Environment
- Jobs & Careers
- Travel & Hospitality
- Products & Brands
- Encyclopedias & Reference

| Method | $R^{\mathrm{LM}}$ | $P^{\mathrm{LM}}$ | $F_1^{H,LM}$ | $F_1^{\mathrm{LM}}$ |
|---|---|---|---|---|
| Baselines (Flattened) | | | | |
| Q-14B w/ AutoSchemaKG (Bai et al., 2025) | 2.1 | 8.26 | 3.35 | 8.17 |
| Q-14B w/ HippoRAG (Gutiérrez et al., 2024) 2-shot | 8.49 | 32.24 | 13.43 | 16.12 |
| Q-14B flatten | 30.46 | 36.46 | 33.19 | 29.87 |
| Q-32B w/ AutoSchemaKG (Bai et al., 2025) | 2.64 | 11.14 | 4.27 | 9.33 |
| Q-32B w/ HippoRAG (Gutiérrez et al., 2024) 2-shot | 10.12 | 39.7 | 16.13 | 20.03 |
| Q-32B flatten | 28.73 | 37.44 | 32.51 | 29.93 |
| GO-20B w/ AutoSchemaKG (Bai et al., 2025) | 5.57 | 11.96 | 7.6 | 9.26 |
| GO-20B w/ HippoRAG (Gutiérrez et al., 2024) 2-shot | 8.26 | 23.06 | 12.16 | 14.02 |
| GO-20B flatten | 36.94 | 37.88 | 37.40 | 33.61 |
| GO-20B 2-shot flatten | 33.18 | 47.10 | 38.93 | 34.93 |
| GO-120B w/ AutoSchemaKG (Bai et al., 2025) | 6.52 | 16.97 | 9.42 | 12.28 |
| GO-120B w/ HippoRAG (Gutiérrez et al., 2024) 2-shot | 28.57 | 12.12 | 17.02 | 17.22 |
| GO-120B flatten | 36.43 | 34.59 | 35.49 | 31.74 |
| GO-120B 2-shot flatten | 42.27 | 46.26 | 44.18 | 40.40 |
| Baselines (Script-gen) | | | | |
| Q-14B agentic-3-iter | 8.11 | 8.26 | 8.18 | 7.14 |
| Q-14B agentic-3-iter 2-shot | 8.59 | 11.13 | 9.70 | 8.01 |
| Q-32B agentic-3-iter | 10.41 | 9.08 | 9.70 | 8.74 |
| Q-32B agentic-3-iter 2-shot | 18.56 | 27.20 | 22.07 | 19.41 |
| Q-72B agentic-3-iter | 9.67 | 9.65 | 9.66 | 7.19 |
| Q-72B agentic-3-iter 2-shot | 16.40 | 19.41 | 17.78 | 14.97 |
| GO-20B agentic-3-iter | 24.70 | 23.22 | 23.94 | 20.87 |
| GO-20B agentic-3-iter 2-shot | 13.06 | 27.30 | 17.66 | 14.40 |
| GO-120B agentic-3-iter | 27.63 | 24.76 | 26.12 | 23.30 |
| GO-120B agentic-3-iter 2-shot | 33.86 | 40.96 | 37.07 | 34.30 |
| GPT-4o agentic-3-iter | 19.05 | 14.72 | 16.61 | 13.81 |
| GPT-4o agentic-3-iter 2-shot | 25.95 | 33.04 | 29.07 | 24.42 |
| L-70B agentic-3-iter | 10.05 | 15.49 | 12.19 | 10.47 |
| L-70B agentic-3-iter 2-shot | 8.23 | 8.08 | 8.15 | 7.10 |
| SCRIBES | | | | |
| Q-14B | 22.96 | 24.26 | 23.59 | 19.91 |
| Q-14B (+CC) | 25.24 | 22.98 | 24.05 | 21.77 |
| Q-32B | 29.88 | 31.53 | 30.68 | 28.05 |
| Q-32B (+CC) | 37.41 | 36.03 | 36.71 | 33.24 |

Table 11: List of all baselines and SCRIBES-trained models. LLM-judged metrics on all data. $P^{\mathrm{LM}}$, $R^{\mathrm{LM}}$, harmonic $F_1^{H,LM}$, and average per-example $F_1^{\mathrm{LM}}$.

We placed *Products & Brands* and *Encyclopedias & Reference* in the test set, with all remaining categories assigned to the training set. This split yielded 196 training examples and 72 test examples. After applying the maximum-context constraint described in Section D.1, 147 training examples and 59 test examples remained.

### G.4 COMPLETE BASELINE NUMBERS

For $F_1$, we provide two variants: (i) the macro-average of per-example $F_1$ scores, and (ii) a harmonic-mean variant defined as

$$F_1^H = \frac{2\overline{PR}}{\overline{P} + \overline{R}} \tag{4}$$

where $\overline{P}$ and $\overline{R}$ denote the mean precision and recall, respectively. The complete list of baseline performance is shown in Table 11 and 12.

## H DETAILS ON QA DATASET USED IN SEC. 5.1

The QA pairs in Sec. 5.1 are collected by Sun et al. (2025) through an LLM-generated followed by human-auditing process. We summarize their process below:

| Method | Example | | | | Holdout | | | |
|---|---|---|---|---|---|---|---|---|
| | $R^{\text{LM}}$ | $P^{\text{LM}}$ | $F_1^{H,LM}$ | $F_1^{\text{LM}}$ | $R^{\text{LM}}$ | $P^{\text{LM}}$ | $F_1^{H,LM}$ | $F_1^{\text{LM}}$ |
| Baselines | | | | | | | | |
| Q-14B agentic-3-iter | 11.96 | 11.81 | 11.88 | 10.57 | 6.47 | 6.90 | 6.68 | 5.77 |
| Q-14B agentic-3-iter 2-shot | 13.21 | 17.97 | 15.23 | 12.63 | 6.29 | 7.79 | 6.96 | 5.73 |
| Q-32B agentic-3-iter | 18.84 | 17.17 | 17.97 | 16.46 | 6.36 | 5.33 | 5.80 | 5.07 |
| Q-32B agentic-3-iter 2-shot | 24.53 | 34.83 | 28.79 | 25.90 | 15.79 | 23.91 | 19.02 | 16.40 |
| Q-72B agentic-3-iter | 13.03 | 13.15 | 13.09 | 10.12 | 8.20 | 8.12 | 8.16 | 5.94 |
| Q-72B agentic-3-iter 2-shot | 24.11 | 28.59 | 26.16 | 21.78 | 13.26 | 15.83 | 14.43 | 12.38 |
| GO-20B agentic-3-iter | 29.25 | 26.38 | 27.74 | 24.91 | 22.51 | 21.78 | 22.14 | 18.94 |
| GO-20B agentic-3-iter 2-shot | 13.48 | 27.68 | 18.13 | 14.41 | 13.07 | 27.11 | 17.64 | 14.66 |
| GO-120B agentic-3-iter | 31.32 | 26.76 | 28.86 | 25.70 | 25.86 | 23.86 | 24.82 | 22.16 |
| GO-120B agentic-3-iter 2-shot | 35.83 | 42.27 | 38.78 | 36.60 | 32.98 | 40.47 | 36.34 | 33.26 |
| GPT-4o agentic-3-iter | 25.19 | 18.35 | 21.23 | 18.47 | 16.00 | 12.89 | 14.28 | 11.47 |
| GPT-4o agentic-3-iter 2-shot | 32.98 | 36.48 | 34.64 | 31.19 | 22.52 | 31.32 | 26.20 | 21.11 |
| L-70B agentic-3-iter | 16.65 | 23.76 | 19.58 | 16.78 | 6.86 | 11.16 | 8.49 | 7.36 |
| L-70B agentic-3-iter 2-shot | 7.77 | 6.77 | 7.23 | 6.18 | 8.42 | 8.68 | 8.54 | 7.51 |
| SCRIBES | | | | | | | | |
| Q-14B | 31.22 | 29.81 | 30.50 | 26.71 | 19.01 | 21.65 | 20.24 | 16.66 |
| Q-14B (+CC) | 34.88 | 30.96 | 32.80 | 29.96 | 20.45 | 19.06 | 19.73 | 17.69 |
| Q-32B | 31.99 | 33.88 | 32.90 | 30.32 | 28.79 | 30.28 | 29.51 | 26.83 |
| Q-32B (+CC) | 39.54 | 35.48 | 37.40 | 34.60 | 36.24 | 36.15 | 36.20 | 32.41 |

Table 12: List of all baselines and SCRIBES-trained models by Example and Holdout. LLM-judged metrics on all data. $P^{\text{LM}}$, $R^{\text{LM}}$, harmonic $F_1^{H,LM}$, and average per-example $F_1^{\text{LM}}$.

First, a 70B Llama model generated initial question–answer pairs using webpage content and ground truth data. Then, these pairs were refined by:

1. **Removing overly complex questions** that required heavy reasoning, focusing instead on comprehension of semi-structured webpages.

2. **Eliminating compound questions** that combined multiple queries into one to avoid inflated difficulty.

3. **Filtering out trivial questions** that all tested models answered correctly, ensuring better differentiation among model performances.

Finally, **human auditors** reviewed and removed any pairs that were ungrounded in the source content or contained incorrect answers.

For more details and statistics, refer to Sun et al. (2025).

## I   EXAMPLE MODEL PREDICTIONS

Here, we show an example of our best model's prediction on the eval set to better illustrate the task and associated difficulties.

For this website, our model generated the code shown in Table 13. The prediction versus gold entries are shown in Table 14. This code achieves the following scores: $R^{\text{LM}} = 53.3$, $P^{\text{LM}} = 50.0$, $F_1^{\text{LM}} = 51.6$.

The model code misses some information such as the "Business Address" and "Mailing Address" fields, penalizing the recalls. It also produces some false positives, including triples where the relation is "sequence" but no corresponding object is returned, reducing precision. Additionally, the model incorrectly treats the filing document number as the subject throughout, rather than using more appropriate subjects such as "Class/Contract C000008114."

```python
from bs4 import BeautifulSoup
from typing import List

def main(html: str) -> List[List[str]]:
    soup = BeautifulSoup(html, 'html.parser')
    triples = []

    # Extract title
    title = soup.title.text if soup.title else ""
    triples.append(["EDGAR Filing", "title", title])

    # Extract form information
    form_name = soup.find("div", id="formName")
    if form_name:
        triples.append([title, "form", form_name.text.strip()])

    sec_num = soup.find("div", id="secNum")
    if sec_num:
        triples.append([title, "accessionNumber", sec_num.text.strip()])

    # Extract filing details
    info_heads = soup.find_all("div", class_="infoHead")
    info_values = soup.find_all("div", class_="info")
    for head, value in zip(info_heads, info_values):
        triples.append([title, head.text.strip(), value.text.strip()])

    # Extract document table information
    table = soup.find("table", class_="tableFile")
    if table:
        rows = table.find_all("tr")
        for row in rows[1:]:  # Skip header row
            cols = row.find_all(["td", "th"])
            if len(cols) >= 5:
                seq = cols[0].text.strip()
                description = cols[1].text.strip()
                document = cols[2].text.strip()
                doc_type = cols[3].text.strip()
                size = cols[4].text.strip()
                triples.append([title, "sequence", seq])
                triples.append([title, "description", description])
                triples.append([title, "document", document])
                triples.append([title, "type", doc_type])
                triples.append([title, "size", size])

    # Extract series and classes information
    series_table = soup.find("table", class_="tableSeries")
    if series_table:
        rows = series_table.find_all("tr")
        for row in rows[2:]:  # Skip header rows
            cols = row.find_all(["td", "th"])
            if len(cols) >= 4:
                status = cols[0].text.strip()
                name = cols[2].text.strip()
                ticker = cols[3].text.strip()
                triples.append([title, "status", status])
                triples.append([title, "name", name])
                triples.append([title, "tickerSymbol", ticker])

    return triples
```

Table 13: An example SCRIBES-generated code discussed in Appendix I.

| Subject | Relation | Object |
|---|---|---|
| **Gold** | | |
| EDGAR Filing Documents for 0000894189-10-002875 | Filing Date | 2010-08-10 |
| EDGAR Filing Documents for 0000894189-10-002875 | Accepted | 2010-08-10 10:58:25 |
| EDGAR Filing Documents for 0000894189-10-002875 | Documents | 2 |
| EDGAR Filing Documents for 0000894189-10-002875 | Period of Report | 2010-06-30 |
| EDGAR Filing Documents for 0000894189-10-002875 | Effectiveness Date | 2010-08-10 |
| empiric_63010nq.htm | Seq | 1 |
| empiric_63010nq.htm | Description | QUARTERLY NOTICE OF PORTFOLIO HOLDINGS |
| empiric_63010nq.htm | Type | N-Q |
| empiric_63010nq.htm | Size | 530257 |
| certs.htm | Seq | 2 |
| certs.htm | Description | OFFICER CERTIFICATIONS |
| certs.htm | Type | EX-99.CERT |
| certs.htm | Size | 24222 |
| 0000894189-10-002875.txt | Description | Complete submission text file |
| 0000894189-10-002875.txt | Size | 556496 |
| Series S000002964 | Name | Core Equity Fund |
| Class/Contract C000008114 | Name | C |
| Class/Contract C000008114 | Ticker Symbol | EMCCX |
| Class/Contract C000008115 | Name | A |
| Class/Contract C000008115 | Ticker Symbol | EMCAX |
| EDGAR Filing Documents for 0000894189-10-002875 | EMPIRIC FUNDS, INC (Filer) CIK | 0001000069 (see all company filings) |
| EDGAR Filing Documents for 0000894189-10-002875 | IRS No. | 742759654 |
| EDGAR Filing Documents for 0000894189-10-002875 | State of Incorp. | TX |
| EDGAR Filing Documents for 0000894189-10-002875 | Fiscal Year End | 931 |
| EDGAR Filing Documents for 0000894189-10-002875 | Type | N-Q |
| EDGAR Filing Documents for 0000894189-10-002875 | Act | 40 |
| EDGAR Filing Documents for 0000894189-10-002875 | File No. | 811-09088 |
| EDGAR Filing Documents for 0000894189-10-002875 | Film No. | 101003905 |
| EDGAR Filing Documents for 0000894189-10-002875 | Business Address | 6300 BRIDGEPOINT PARKWAY BUILDING II, SUITE 105 AUSTIN TX 78730 5123289321X1 |
| EDGAR Filing Documents for 0000894189-10-002875 | Mailing Address | 6300 BRIDGEPOINT PARKWAY BUILDING II, SUITE 105 AUSTIN TX 78730 |
| **Predicted** | | |
| EDGAR Filing | title | EDGAR Filing Documents for 0000894189-10-002875 |
| EDGAR Filing Documents for 0000894189-10-002875 | form | Form N-Q - Quarterly Schedule of portfolio holdings of management investment companies: |
| EDGAR Filing Documents for 0000894189-10-002875 | accessionNumber | SEC Accession No. 0000894189-10-002875 |
| EDGAR Filing Documents for 0000894189-10-002875 | Filing Date | 2010-08-10 |
| EDGAR Filing Documents for 0000894189-10-002875 | Accepted | 2010-08-10 10:58:25 |
| EDGAR Filing Documents for 0000894189-10-002875 | Documents | 2 |
| EDGAR Filing Documents for 0000894189-10-002875 | Period of Report | 2010-06-30 |
| EDGAR Filing Documents for 0000894189-10-002875 | Effectiveness Date | 2010-08-10 |
| EDGAR Filing Documents for 0000894189-10-002875 | sequence | 1 |
| EDGAR Filing Documents for 0000894189-10-002875 | description | QUARTERLY NOTICE OF PORTFOLIO HOLDINGS |
| EDGAR Filing Documents for 0000894189-10-002875 | document | empiric_63010nq.htm |
| EDGAR Filing Documents for 0000894189-10-002875 | type | N-Q |
| EDGAR Filing Documents for 0000894189-10-002875 | size | 530257 |
| EDGAR Filing Documents for 0000894189-10-002875 | sequence | 2 |
| EDGAR Filing Documents for 0000894189-10-002875 | description | OFFICER CERTIFICATIONS |
| EDGAR Filing Documents for 0000894189-10-002875 | document | certs.htm |
| EDGAR Filing Documents for 0000894189-10-002875 | type | EX-99.CERT |
| EDGAR Filing Documents for 0000894189-10-002875 | size | 24222 |
| EDGAR Filing Documents for 0000894189-10-002875 | sequence | |
| EDGAR Filing Documents for 0000894189-10-002875 | description | Complete submission text file |
| EDGAR Filing Documents for 0000894189-10-002875 | document | 0000894189-10-002875.txt |
| EDGAR Filing Documents for 0000894189-10-002875 | type | |
| EDGAR Filing Documents for 0000894189-10-002875 | size | 556496 |
| EDGAR Filing Documents for 0000894189-10-002875 | status | CIK 0001000069 |
| EDGAR Filing Documents for 0000894189-10-002875 | name | |
| EDGAR Filing Documents for 0000894189-10-002875 | tickerSymbol | |
| EDGAR Filing Documents for 0000894189-10-002875 | status | Class/Contract C000008114 |
| EDGAR Filing Documents for 0000894189-10-002875 | name | C |
| EDGAR Filing Documents for 0000894189-10-002875 | tickerSymbol | EMCCX |
| EDGAR Filing Documents for 0000894189-10-002875 | status | Class/Contract C000008115 |
| EDGAR Filing Documents for 0000894189-10-002875 | name | A |
| EDGAR Filing Documents for 0000894189-10-002875 | tickerSymbol | EMCAX |

Table 14: Comparison of predicted and gold triples for Code 13.

## J   PROMPTS USED

All prompts used in our experiments are shown here in Jinja2 format, including the classifier prompt (Prompt 15), LLM direct extraction prompt (Prompt 16), LLM-as-a-judge prompt (Prompt 17), QA prompt (Prompt 18), the main script generation prompt (Prompt 19) used in both baseline and in SCRIBES training data, and the QA evaluation prompt (Prompt 20).

```
# instruction

Your task is to classify an input HTML to see whether it contains semi-structured content.

You are shown below with one example with semi-structured content and one without.
Output a JSON with the following two fields: "reason" and "decision".
Reason should specify your chain of thought and decision should be one of:

- Semi-structured content: Respond with "Yes" if the HTML contains semi-structured content,
such as tables and infoboxes.
- No semi-structured content: Respond with "No" if the HTML does not contain any semi-structured content.
- Explicit content: Respond with "Exclude" if the HTML contains explicit content
(e.g., adult material, graphic violence).

# input

Exaples containing the following HTML:

{{ HTML_example_1 }}

# output

{
    "reason": "This HTML contains a table which falls into the definition of semi-structured content",
    "decision": "Yes"
}

# input

{{ HTML_example_2 }}

# output

{
    "reason": "Even though this HTML contains structured discussions and Q&As, it does not have tables or
        infoboxes",
    "decision": "No"
}

# input

An HTML with the following info:

{{ HTML_example_3 }}

# output

{
    "reason": "This HTML show cases a infobox, which should be treated as a semi-structured content.",
    "decision": "Yes"
}

# input

{{ html }}
```

Table 15: Classifier prompt used to determine whether a webpage contains semi-structured content or not.

```
# instruction
You are given a doc in HTML and its title. Please return all (subject, predicate, object) triples
that can be extracted from the doc, in the order they appear in the doc. For large chunk of descriptions
or sections of free-form text, you should keep them as object. Do not attempt to break big chunks
of texts down into smaller portions.

Subject, predicate, and object should generally be gained from the text spans in the doc or the title.
Please only include complete triples; if for any section the predicate or object is missing from the doc,
you may skip it.
Output a list of lists, where each inner list is a triple. I will use python's eval to parse your output.

# input
{% if example_global_html_triples %}
Here are {{ example_global_html_triples|length }} examples of flattened HTML pages and their expected triples:
{% for single_example in example_global_html_triples %}
Example {{ loop.index0 }} Flattened HTML: {{ single_example["html_flatten"] }}
Example {{ loop.index0 }} Expected Triples: {{ single_example["triples_annotation"] }}
{% endfor %}
{% endif %}

{% if example_triples %}
Here are 10 triples we are expecting in the output randomly chosen: {{ example_triples }}
{% endif %}

### title
{{ html_title }}

### HTML
{{ html }}
```

Table 16: LLM direct extraction prompt used to directly generate triples from a webpage.

```
# instruction

You are given two (subject, predicate, object) triples.
Your response should be "Yes" if the triples are semantically the same or "No"
if they are semantically different.

# input
{{ tx }}
{{ ty }}
```

Table 17: LLM-as-a-judge prompt for judging whether two triples are semantically equivalent.

```
# instruction
You are given a question and a reference that may or may not help answer the question.
Please answer the question. Be concise.

# input
### Question
{{ question }}
### Reference
{{ reference }}
```

Table 18: Question Answering prompt with reference.

```
# instruction

Your task is to generate semantic triples from a given HTML.
A triple contains a subject, a predicate, and an object.
You should write python code to extract triples from the HTML.
The final executable function should be called `def main(html) -> List[tuple(str, str, str)]:`,
where it will output a list of triples.
You should output the python code only. Feel free to add comments to explain your code.
Do not include any text other than the code in your response.

IMPORTANT: we will re-use the same script for other webpages with similar HTML contents.
So you should make your script re-usable across different websites
(do not hardcode for values for this particular HTML).

# input

{% if example_global_html_triples %}
Here are {{ example_global_html_triples|length }} examples of other HTML sites and
what the script-generated output we are looking for:
{% for single_example in example_global_html_triples %}
Example {{ loop.index0 }} HTML: {{ single_example["html_content"] }}
Example {{ loop.index0 }} Expected Outputs: {{ single_example["triples_annotation"] }}
{% endfor %}
{% endif %}

HTML: {{ html }}

{% if example_triples %}
Here are 10 triples we are expecting in the output randomly chosen: {{ example_triples }}
{% endif %}
{% if all_triples %}
Here are all the triples we are expecting in the output: {{ all_triples }}
{% endif %}

{% if prev_script %}
You previously generated a script:
{{ prev_script }}

This script generated the following result:
{{ feedback }}

If you think the results are good enough, stop and output the same script.
If not, incorporate the feedback in generating a new script.
{% endif %}
```

Table 19: Main script generation prompt for baselines and SCRIBES-trained models.

```
# instruction

You need to check whether the prediction of a question-answering system to a question is correct.
You should make the judgment based on the ground truth answer provided to you.

Your response should be "correct" if the prediction is correct or "incorrect" if the prediction is wrong.

# input

Question: {{ question }}
Ground truth: {{ gold }}
Prediction: {{ answer }}
Correctness:
```

Table 20: QA evaluation prompt.

