# OpenReview forum: "SCRIBES: Web-Scale Script-Based Semi-Structured Data Extraction with Reinforcement Learning"
_ICLR.cc/2026/Conference — ICLR 2026 Poster_

### Official Review · Reviewer_TGsp · 2025-10-23

**Soundness:** 2
**Presentation:** 2
**Contribution:** 2
**Rating:** 4
**Confidence:** 4

**Summary:**

The paper introduces SCRIBES, a reinforcement learning (RL) framework for scalable information extraction from semi-structured web data (HTML tables, lists, infoboxes). Rather than invoking a large language model (LLM) for each page individually, SCRIBES trains an LLM to generate reusable extraction scripts that can be applied to groups of structurally similar webpages (e.g., pages from the same site).

**Strengths:**

The script-generation idea bridges wrapper induction and LLM-based extraction, offering a path to web-scale efficiency.

The use of cross-page generalization as a verifiable RL reward is elegant and practical.

The empirical results also show some promising evidence.

**Weaknesses:**

The system largely combines existing ingredients (RLVR, GRPO, LLM-as-a-judge, fuzzy matching) rather than introducing a fundamentally new algorithm. This is my biggest concern to give a positive score. The “fuzzy F1” and LLM-judged metrics may still be noisy; no human verification of RL reward quality was reported. In addition, although SCRIBES aims for efficiency, it still relies on large proprietary models (e.g., GPT-4o, Qwen-32B) for training and reward calculation, limiting replicability and true scalability.

**Questions:**

How sensitive is performance to noise in synthetic LLM-based rewards? Any quantitative correlation between F_fuzzy and F_LM?

Does a model trained on finance/legal tables transfer to product or encyclopedia pages?

What is the total compute cost (training + inference) compared to a standard per-page LLM pipeline?

Could you show examples where the generated scripts fail catastrophically (e.g., mismatched tags or wrong data fields)?

---

> ### Author Response · Authors · 2025-11-25
>
> We’d like to thank the reviewer for your insightful criticism and feedback! Below, we will address your concerns point-by-point. We have also updated the paper with new experiments highlighted in red and will also include results in our response below:
>
> > The system largely combines existing ingredients (RLVR, GRPO, LLM-as-a-judge, fuzzy matching) rather than introducing a fundamentally new algorithm.
>
> While SCRIBES indeed builds upon several existing components, we emphasize that it introduces several **distinct and substantive innovations** beyond prior work:
>
> 1. **A novel script-based extraction paradigm.** We propose a novel approach that uses script generation to extract semi-structured content. This is, to our knowledge, the first attempt to replace per-page LLM inference with reusable extraction scripts. This paradigm is far more suitable for web-scale extraction, offering improved accuracy compared to traditional information extraction (IE) methods and much better efficiency compared to costly per-page LLM inference.
> 2.  **Novel RL setup and data collection strategy.** Beyond training solely on labeled data, we introduce a new reinforcement learning framework that leverages “in-the-wild” webpages to collect additional supervision signals. By utilizing noisy rewards as synthetic data, our approach demonstrates a practical pathway for scalable, self-improving extraction.
> 3.  **Demonstrated downstream benefits.** We further show that SCRIBES-extracted triples substantially enhance downstream QA performance, even for state-of-the-art models such as GPT-4o, achieving over a **4% improvement** compared to baselines.
>
> In summary, SCRIBES establishes a novel research direction by combining methodological contributions with measurable downstream impact. We believe it opens new opportunities for scalable, structure-aware learning applicable to future tasks such as multi-page, complex QA, and pre-training.
>
> > The “fuzzy F1” and LLM-judged metrics may still be noisy; no human verification of RL reward quality was reported.
>
> Thank you for raising this concern. We followed the same evaluation metrics as defined in the SemiBench paper, which reported a 95% agreement rate between the LLM-based F1 metric and human judgments, indicating strong alignment. We have updated the paper in Appendix F.3 to clarify this point and to explicitly note that our evaluation framework inherits this validated setup.
>
> > In addition, although SCRIBES aims for efficiency, it still relies on large proprietary models (e.g., GPT-4o, Qwen-32B) for training and reward calculation, limiting replicability and true scalability.
>
> First, we clarify that SCRIBES aims for **efficiency for web-scale knowledge extraction**. Our proposed approach is script-based, i.e. generating a script for each web domain using our post-trained LLM and then executing the script on all webpages in the web domain. This method is much more cost-effective than applying LLMs on every webpage. More details can be found in the answer to your question on training compute below.
>
> Additionally, we emphasize that we use mostly open-sourced models (e.g. Qwen-2.5-instruct family and GPT-OSS family) and limit the use of proprietary models such as GPT-4o to only evaluation purposes. All main results in the paper are obtained using the open-source **Qwen-2.5-instruct** models, which are fully open-sourced. Our training pipeline involving CommonCrawl also only makes use of open-sourced models such as GPT-OSS. These design choices ensure that our approach remains **fully replicable** and **scalable** without dependence on proprietary systems.
>
> > How sensitive is performance to noise in synthetic LLM-based rewards? Any quantitative correlation between F_fuzzy and F_LM?
>
> Thank you for raising this important question. As discussed in Section 3.3 and Appendix F of our paper, we do not use $F_1^{\text{LM}}$ as the training reward; instead, we employ $F_1^{\text{fuzzy}}$ as a more stable and computationally efficient proxy. This design choice avoids long latency from LLM-based judgments while still providing reliable reward signals.
>
> We have also updated Appendix F.1 with new information on the correlation between the two metrics:
>
> Empirically, we observe a correlation between $F_1^{\text{fuzzy}}$ and $F_1^{\text{LM}}$. The latter tends to yield slightly lower absolute scores but exhibits the same performance trend across models and configurations. A comparison showing the two metrics and the associated precision and recall metrics for the baselines are shown in Table below. We calculated the correlation coefficient between $F_1^{\text{fuzzy}}$ and $F_1^{\text{LM}}$ to be $0.957$ with a p-value of $1.4 \times 10^{-5}$, showing a strong correlation.

---

> ### Author Response · Authors · 2025-11-25
>
> | **Method**              | $R^{\mathrm{LM}}$ | $P^{\mathrm{LM}}$ | $F_1^{\mathrm{LM}}$ | $R^{\mathrm{fuzzy}}$ | $P^{\mathrm{fuzzy}}$ | $F_1^{\mathrm{fuzzy}}$ |
> |-------------------------|-------------------|-------------------|----------------------|-----------------------|-----------------------|-------------------------|
> | Q-14B flatten           | 30.46 | 36.46 | 29.87 | 45.96 | 52.37 | 43.50 |
> | Q-32B flatten           | 28.73 | 37.44 | 29.93 | 41.62 | 54.25 | 42.26 |
> | GO-20B 2-shot flatten   | 33.18 | 47.10 | 34.93 | 46.53 | 65.21 | 49.77 |
> | GO-120B 2-shot flatten  | 42.27 | 46.26 | 40.40 | 56.01 | 61.42 | 53.37 |
> | Q-14B 3-iter 2-shot     | 8.59  | 11.13 | 8.01  | 17.17 | 25.57 | 16.53 |
> | Q-72B 3-iter 2-shot     | 16.40 | 19.41 | 14.97 | 28.73 | 37.96 | 28.60 |
> | Q-32B 3-iter 2-shot     | 18.56 | 27.20 | 19.41 | 27.49 | 44.67 | 30.39 |
> | GO-20B 3-iter           | 24.70 | 23.22 | 20.87 | 52.30 | 41.83 | 39.58 |
> | GPT-4o 3-iter 2-shot    | 25.95 | 33.04 | 24.42 | 45.58 | 60.57 | 44.46 |
> | GO-120B 3-iter 2-shot   | 33.86 | 40.96 | 34.30 | 49.79 | 65.72 | 52.02 |
>
>
> > Does a model trained on finance/legal tables transfer to product or encyclopedia pages?
>
> Thank you for asking this interesting question. To study this, we have conducted an additional ablation study in our updated Section 4.4 Ablations. Here is the relevant section and findings:
>
> **RQ6**: Do SCRIBES-trained models transfer across domains? For example, does a model trained on finance or legal tables generalize to product or encyclopedia pages?
>
> To investigate this question, we conduct an ablation study using a train–test split in which the test set contains all product and encyclopedia pages, while the training set excludes webpages from these domains entirely. Details on this setup are provided in Appendix G.3. As shown in Table below, the SCRIBES-trained model still substantially outperforms the strongest agentic baseline of the same model by more than 10%. To develop a model capable of web-scale extraction, we recommend training on a dataset that encompasses diverse domains and page layouts, as demonstrated by our CommonCrawl processing in Section 3.3.2.
>
> | **Model and Method**           | $R^{\mathrm{LM}}$ (All) | $P^{\mathrm{LM}}$ (All) | $F_1^{\mathrm{LM}}$ (All) | $R^{\mathrm{LM}}$ (Example) | $P^{\mathrm{LM}}$ (Example) | $F_1^{\mathrm{LM}}$ (Example) | $R^{\mathrm{LM}}$ (Holdout) | $P^{\mathrm{LM}}$ (Holdout) | $F_1^{\mathrm{LM}}$ (Holdout) |
> |--------------------------------|--------------------------|--------------------------|----------------------------|------------------------------|------------------------------|-------------------------------|------------------------------|------------------------------|-------------------------------|
> | Q-14B agentic 3-iter 2-shot    | 9.5  | 13.7 | 8.8  | 23.2 | 24.2 | 20.0 | 12.4 | 7.4  | 7.2  |
> | Q-14B (SCRIBES)              | **20.7** | **22.2** | **19.4** | **31.8** | **36.1** | **30.4** | **14.5** | **12.0** | **12.2** |
>
> In this ablation study, we reorganized the dataset by assigning each website to one of the following content categories:
>
> - Finance \& Economics
> - Legal \& Regulatory
> - Developer \& Software
> - Science \& Research
> - Science \& Database
> - Sports
> - Gaming \& Entertainment
> - Media \& Entertainment
> - Real Estate
> - Social Platforms
> - Weather \& Environment
> - Jobs \& Careers
> - Travel \& Hospitality
> -  Products \& Brands
> - Encyclopedias \& Reference
>
> We placed *Products \& Brands* and *Encyclopedias \& Reference* in the test set, with all remaining categories assigned to the training set.
>
> > What is the total compute cost (training + inference) compared to a standard per-page LLM pipeline?
>
> We have updated our manuscript with calculations on this in Section 4.3, Figure 4, and Appendix D.3. Here is the relevant section:
>
> We further compare the total GPU cost of SCRIBES, including training, with per-page LLM inference in Figure 4. Let $g$ denote the number of groups processed. While per-page inference (dashed purple line) increases linearly with both the number of groups $g$ and the group size $k$, the SCRIBES-trained model yields substantial FLOP savings, with the magnitude of savings growing proportionally to group size. For instance, with 100 webpages per website group, SCRIBES can already provide a computational saving of $1.12 \times 10^{21}$ FLOPS when processing $10^5$ groups. Additional details on the FLOP estimates are provided in  Appendix D.3.
>
> > Could you show examples where the generated scripts fail catastrophically (e.g., mismatched tags or wrong data fields)?
>
> Thank you for this suggestion. We have updated the manuscript with such examples in Appendix I.

---

### Official Review · Reviewer_vXwT · 2025-10-30

**Soundness:** 3
**Presentation:** 3
**Contribution:** 3
**Rating:** 6
**Confidence:** 3

**Summary:**

In this paper, the authors target the problem of semi-structured data extraction.
They develop a script-based semi-structured content extraction at web-scale framework to improve generalization of this task. The proposed reinforcement learning framework leverages layout similarity across webpages within the same
site as the reward signal. The proposed approach outperforms strong baselines in script quality and boosts downstream question answering accuracy.

**Strengths:**

1. The paper addresses an important and practical problem:  extracting structured data from the vast amounts of semi‑structured web content.
2. The framework is novel in that it focuses on webpage grouping and employs a semi-supervised-like approach, enhancing the model's ability to generate highly reusable scripts, thereby improving task performance while reducing costs.
3. The experimental results reveal that the designed framework is efficient and effective in both generating robust, reusable scripts and improving performace on downstream tasks.

**Weaknesses:**

1. The advantages of script-based extraction need to be further elaborated.
2. The domain scalability of the method requires further elaboration.
3. The evaluation baselines seem to be somehow weak.

**Questions:**

1. A "script" is essentially akin to a type of automatically generable rule, yet the paper does not clearly distinguish it from traditional wrappers, templates, or XPath rules.
There is a potential risk of overstating the method's novelty. It is necessary to more explicitly articulate the advantages of RL-generated scripts—such as generalization ability and automatic learning strategies—highlighting their unique value compared to manually crafted rules.

2. The paper emphasizes black-box data analysis while providing limited discussion on the limitations of the DSL and the handling of anomalous webpages. Its capability to adapt to irregular structures—such as multi-row/multi-column tables, nested lists, or JavaScript-rendered content—remains unclear.

3. The paper claims support for "web-scale" and multi-domain scenarios, yet it does not present the domain differences between the webpages used for training and those used for testing. Including such statistics would make the experimental conclusions more convincing.

4. The selected baselines appear relatively weak. Many other LLM-based methods are mentioned in Section 1; it remains unclear how the proposed method performs in comparison with them.

---

> ### Author Response · Authors · 2025-11-25
>
> We’d like to thank the reviewer for your insightful feedback! Below, we will address your concerns point-by-point. We have also updated the paper with new experiments highlighted in red, and we will also include results in our response below:
>
> > A "script" is essentially akin to a type of automatically generable rule, yet the paper does not clearly distinguish it from traditional wrappers, templates, or XPath rules. There is a potential risk of overstating the method's novelty. It is necessary to more explicitly articulate the advantages of RL-generated scripts—such as generalization ability and automatic learning strategies—highlighting their unique value compared to manually crafted rules.
>
> Thank you for this comment! We have updated our Section 2 to better contrast with existing IE strategies:
>
> Traditional rule-based approaches typically rely on manually designed heuristics that locate syntactically plausible spans (e.g., via template matching), but they lack the ability to interpret webpage semantics or reason over structural variability. In contrast, our approach learns to write customized executable scripts (i.e. full extraction programs that operate directly on raw HTML) based on semantic understanding of the webpages. This enables the system to generalize beyond fixed rules and adapt automatically without manual template design.
>
>
> > The paper emphasizes black-box data analysis while providing limited discussion on the limitations of the DSL and the handling of anomalous webpages. Its capability to adapt to irregular structures—such as multi-row/multi-column tables, nested lists, or JavaScript-rendered content—remains unclear.
>
> Thank you for this suggestion. Following your feedback, we have conducted additional error analysis in our updated Section 4.5. Here is the relevant section:
>
> We also compare the performance of our model's outputs on contents involving multi-column and nested lists. As shown in Table below, we observe that such content is more challenging for our model. Further prediction examples are showcased in Appendix I.
>
>
> | **Breakdown**    | $P^{\mathrm{LM}}$ | $R^{\mathrm{LM}}$ | $F_1^{\mathrm{LM}}$ |
> |------------------|----------------------|----------------------|------------------------|
> | Nested list      | 23.6 | 18.7 | 19.7 |
> | Multi-column     | 23.3 | 33.9 | 18.6 |
> | **All**          | **39.5** | **35.5** | **34.6** |
>
> **Table 5b: Comparison of the best Q-32 model’s performance across nested-list and multi-column cases on our test set.** A *nested list* is defined as content containing a table or another list embedded within an outer list.
> *Multi-column content* refers to cases where headers span multiple columns.
>
> Regarding JavaScript-rendered content, the performance primarily depends on the HTML collection process. When JavaScript content is rendered during collection (e.g., via tools such as Selenium), the resulting HTML structure is largely equivalent to non-JavaScript pages, and thus SCRIBES can handle them without substantial differences.

---

> ### Author Response · Authors · 2025-11-25
>
> > The paper claims support for "web-scale" and multi-domain scenarios, yet it does not present the domain differences between the webpages used for training and those used for testing. Including such statistics would make the experimental conclusions more convincing.
>
> Thank you for this great suggestion. We have completed additional statistics computation in Appendix E and Table 8. Here is the relevant snippet:
>
> We compare several statistics of HTML webpages in the table below. We define each statistic:
> -   DOM Max Depth: The maximum depth of the Document Object Model (DOM) tree in an HTML document. This measures how deeply the elements are nested; a higher DOM Max Depth indicates more extensive nesting.
> -   Deduplication Ratio: The lengths of the HTML content before and after applying the deduplication algorithm described in Appendix C (in characters). This quantifies redundancy in the HTML structure; a lower Deduplication Ratio indicates greater redundancy.
> -   Structure Ratio: The ratio of the HTML length to the flattened text length (in characters). This approximates how much structural markup the HTML contains relative to its textual content; a higher Structure Ratio reflects more structural complexity.
> -   Tag Count: The number of all tags in an HTML document. This measures the structural complexity of the HTML; a higher Tag Count indicates a more complex document.
>
>
> | **Feature** | **Metric** | **Train** | **Test** | **CC (After Step 6 in Fig. 3)** |
> |-------------|------------|-----------|-----------|-----------------------------------------------------|
> | **DOM Max Depth** | Mean   | 20.2  | 18.4  | 20.2  |
> |               | Median | 19.0  | 17.0  | 17.0  |
> |               | Std    | 4.94  | 6.98  | 21.7  |
> |               | Min    | 10.0  | 10.0  | 5.00  |
> |               | Max    | 37.0  | 37.0  | 455   |
> | **Deduplication Ratio** | Mean   | 0.215  | 0.174  | 0.353  |
> |               | Median | 0.213  | 0.166  | 0.344  |
> |               | Std    | 0.111  | 0.0986 | 0.178  |
> |               | Min    | 0.0302 | 0.0324 | 0.000480 |
> |               | Max    | 0.484  | 0.422  | 1.02   |
> | **Structure Ratio** | Mean   | 46.1  | 43.5  | 20.8  |
> |              | Median | 28.7  | 27.4  | 13.8  |
> |              | Std    | 40.0  | 45.5  | 48.7  |
> |              | Min    | 2.24  | 6.00  | 1.19  |
> |              | Max    | 174   | 199   | 1960  |
> | **Tag Count** | Mean   | 1650  | 1820  | 655   |
> |             | Median | 1260  | 1080  | 496   |
> |             | Std    | 2140  | 2550  | 559   |
> |             | Min    | 224   | 154   | 18.0  |
> |             | Max    | 27800 | 12300 | 5070  |
>
>
> This comparison shows that the labeled training and test sets share similar summary statistics, whereas the CommonCrawl portion differs noticeably. In particular, the CommonCrawl data is less redundant (lower Deduplication Ratio), contains less structural markup (lower Structure Ratio), and is structurally simpler (lower Tag Count). Across all metrics, it also exhibits greater variability, as indicated by the higher standard deviations. These observations suggest that incorporating this portion of the CommonCrawl data into training can meaningfully broaden the distribution of inputs, exposing the models to examples that differ substantially from those in the labeled dataset.
>
> > The selected baselines appear relatively weak. Many other LLM-based methods are mentioned in Section 1; it remains unclear how the proposed method performs in comparison with them.
>
> Thank you for raising this important point. We have conducted new experiments incorporating two recent, SOTA LLM-based agentic systems for knowledge base construction: HippoRAG and AutoSchemKG. The corresponding results are presented in the updated Table 1, Table 10, and Appendix G.1. The relevant section is included below for reference:
>
> In addition to the simple 2-shot baseline, we profile two SOTA LLM-based agentic knowledge-base–construction baselines: HippoRAG [1] and AutoSchemaKG [2], representative of recent LLM-driven KG construction pipelines.
>
> HippoRAG is a retrieval-augmented generation framework that builds a knowledge graph as an embedding index, mimicking the role of the hippocampus in human memory. We use the first stage of their KG construction pipeline, which consists of two prompts, one for named entity recognition (NER) and one for triple extraction. We also replace their 1-shot example with the same 2-shot examples used in our baseline.
>
> AutoSchemaKG is a framework for web-scale KG construction over a pretraining-scale corpus. It calls three LLM modules on each webpage: (1) an entity–entity relationship extractor, (2) an entity–event relationship extractor, and (3) an event–event relationship extractor. These prompts are all zero-shot and are challenging to adapt, so we retain them as originally specified.

---

> ### Author Response · Authors · 2025-11-25
>
> As shown in Table below, the simple 2-shot baseline outperform both LLM-based baselines across all models evaluated on our task, including by more than 20% for the strongest model, GPT-OSS-120B. Moreover, they inherit the same cost inefficiencies, as each webpage requires multiple LLM calls.
>
> | Method | $R^{\mathrm{LM}}$ | $P^{\mathrm{LM}}$ | $F_1^{H, LM}$ | $F_1^{\mathrm{LM}}$ |
> |--------|----------------------|----------------------|----------------------|------------------------|
> | **Baselines (Direct LLM Extraction)** | | | |
> | *Q-14B w/ AutoSchemaKG* | 2.1 | 8.26 | 3.35 | 8.17 |
> | *Q-14B w/ HippoRAG (2-shot)* | 8.49 | 32.24 | 13.43 | 16.12 |
> | Q-14B flatten | 30.46 | 36.46 | 33.19 | 29.87 |
> | *Q-32B w/ AutoSchemaKG* | 2.64 | 11.14 | 4.27 | 9.33 |
> | *Q-32B w/ HippoRAG (2-shot)* | 10.12 | 39.70 | 16.13 | 20.03 |
> | Q-32B flatten | 28.73 | 37.44 | 32.51 | 29.93 |
> | *GO-20B w/ AutoSchemaKG* | 5.57 | 11.96 | 7.60 | 9.26 |
> | *GO-20B w/ HippoRAG (2-shot)* | 8.26 | 23.06 | 12.16 | 14.02 |
> | GO-20B flatten | 36.94 | 37.88 | 37.40 | 33.61 |
> | GO-20B 2-shot flatten | 33.18 | 47.10 | 38.93 | 34.93 |
> | *GO-120B w/ AutoSchemaKG* | 6.52 | 16.97 | 9.42 | 12.28 |
> | *GO-120B w/ HippoRAG (2-shot)* | 28.57 | 12.12 | 17.02 | 17.22 |
> | GO-120B flatten | 36.43 | 34.59 | 35.49 | 31.74 |
> | GO-120B 2-shot flatten | 42.27 | 46.26 | 44.18 | 40.40 |
>
>
> References:
>
> [1] Bernal Jimenez Gutierrez, Yiheng Shu, Yu Gu, Michihiro Yasunaga, and Yu Su. Hipporag: Neuro-biologically inspired long-term memory for large language models. In The Thirty-eighth Annual Conference on Neural Information Processing Systems, 2024. URL https://openreview.net/forum?id=hkujvAPVsg.
>
> [2] Jiaxin Bai, Wei Fan, Qi Hu, Qing Zong, Chunyang Li, Hong Ting Tsang, Hongyu Luo, Yauwai Yim, Haoyu Huang, Xiao Zhou, Feng Qin, Tianshi Zheng, Xi Peng, Xin Yao, Huiwen Yang, Leijie Wu, Yi Ji, Gong Zhang, Renhai Chen, and Yangqiu Song. Autoschemakg: Autonomous knowledge graph construction through dynamic schema induction from web-scale corpora, 2025. URL https://arxiv.org/abs/2505.23628.

---

### Official Review · Reviewer_yg8p · 2025-10-31

**Soundness:** 3
**Presentation:** 3
**Contribution:** 2
**Rating:** 6
**Confidence:** 4

**Summary:**

The paper proposes a web-scale structured information extraction framework called SCRIBES. Instead of calling a large model on every single webpage, the key idea is to first let an LLM generate a reusable extraction program (a Python script) and then apply this script to many structurally similar pages. To make the script generalize, the authors use reinforcement learning on a group of webpages to produce scripts that work across pages. Experiments show that, compared to existing script-generation / agentic baselines, their RL-based training improves group-level reusability, and the extracted triples can also benefit downstream HTML QA.

**Strengths:**

* Well-motivated problem. Web-scale information extraction is an important and realistic task, and the cost of applying an LLM to every single page is prohibitive; reducing LLM calls is therefore a meaningful objective.
* Novel paradigm. By generating a script for each layout type (i.e., each group of structurally similar webpages), the method reframes the task from page-level IE to program induction and reuse, clearly setting it apart from pure agent-style, one-page-at-a-time extraction.

**Weaknesses:**

* Absolute performance gap. Despite gains over script-generation baselines at comparable model sizes, the approach still trails the strongest page-wise direct LLM extraction (e.g., GO-120B with flattening).
* Fragile grouping assumption. The heuristic “same domain + similar URL prefix ⇒ similar structure” may not hold on sites with mixed templates, limiting robustness and generalization.
* Bias toward structure-heavy pages. Performance degrades on free-form, text-heavy pages that require deeper semantic understanding.
* Insufficient deployment analysis. A clearer cost–accuracy trade-off (e.g., a curve or table) is needed to show when the script-based approach is preferable to per-page LLM extraction.

**Questions:**

* Could you include one or two representative examples of the generated Python extraction scripts in the appendix , so that readers can better understand what the model actually produces?
* Is it feasible to integrate a lightweight Open IE (or similar semantic post-processing) component into the generated script to improve its ability to handle less strictly structured content?

---

> ### Author Response · Authors · 2025-11-25
>
> We’d like to thank the reviewer for your close read of our paper and insightful comments! Below, we will address your concerns point-by-point. We have also updated the paper with new experiments highlighted in red and will also include results in our response below:
>
> > Absolute performance gap. Despite gains over script-generation baselines at comparable model sizes, the approach still trails the strongest page-wise direct LLM extraction (e.g., GO-120B with flattening).
>
> We believe that **generating scripts to parse knowledge is inherently more challenging than directly extracting triples from flattened HTMLs**, since the latter primarily requires identifying and copying relevant information, whereas the former **additionally** requires understanding similarities between webpages layouts within the same web domain and generating scripts to extract knowledge.
>
> However, **the script-generation approach is much more cost effective.** As we show in Section 4.3, SCRIBES quickly becomes more efficient as long as the target website contains at least 4 structurally similar pages and grows linearly with the number of structurally similar pages.
>
> Additionally, **script-based knowledge extraction has comparable improvement for downstream QA performance** to the much more expensive LLM-based extraction approach, as additional experiment shows in updated Section 5.1. Here is the relevant snippet:
>
> We further observe that although the SCRIBES-trained models slightly underperform the strongest per-page LLM-inference baseline in Table below, they nonetheless deliver comparable downstream QA gains. As shown in the Table below, using SCRIBES-generated triples improves QA performance for Q-14B and GPT-4o, yields roughly similar performance for Q-1.5B and Q-32B, and performs worse for Q-3B and Q-7B. These results indicate that higher IE accuracy does not necessarily translate into better downstream QA performance. Instead, using SCRIBES-produced triples can deliver much better efficiency and a similar level of downstream QA improvement.
>
>
> | **Additional reference**                       | **Q-1.5B** | **Q-3B** | **Q-7B** | **Q-14B** | **Q-32B** | **GPT-4o** |
> |-----------------------------------------------|-----------|----------|----------|-----------|-----------|------------|
> | Flattened HTML                                 | 50.2 | 53.8 | 62.9 | 74.2 | 70.8 | 82.5 |
> | ***+ Best Direct LLM Extraction triples (new)***     | ***52.8*** | ***61.2*** | ***66.6*** | ***73.5*** | ***73.1*** | ***82.7*** |
> | + Best Q-32B triples                           | 52.9 | 54.3 | 64.1 | 77.3 | 73.2 | 86.6 |
> | + Ground truth triples                         | 60.5 | 64.9 | 70.5 | 78.2 | 74.8 | 87.4 |
>
>
> > Fragile grouping assumption. The heuristic “same domain + similar URL prefix ⇒ similar structure” may not hold on sites with mixed templates, limiting robustness and generalization.
>
> We acknowledge that this heuristic is indeed a limiting factor. However, a key contribution of SCRIBES is its ability to substantially narrow the performance gap between the example and holdout sets compared to baseline methods, despite this limitation. We leave the exploration of more robust and scalable heuristic selection strategies for future work.
>
>
> > Bias toward structure-heavy pages. Performance degrades on free-form, text-heavy pages that require deeper semantic understanding.
>
> Our approach is designed for structure-heavy pages where page layout provides rich signals. On text-heavy pages that require deeper semantic understanding, we do not expect the scripts to be able to extract knowledge-rich triplets. For instance, they cannot convert arbitrary free-form text into structured triples using rule-based methods. Such pages are therefore out of scope for this paper, and we leave a more detailed study of free-text performance to future work.
>
>
> > Insufficient deployment analysis. A clearer cost–accuracy trade-off (e.g., a curve or table) is needed to show when the script-based approach is preferable to per-page LLM extraction.
>
> Thank you for this great suggestion! We have updated the paper with a new Figure 4 showing the performance improvement of SCRIBES over per-page LLM inference baselines at web-scale. Here is the relevant section:
>
> We further compare the total GPU cost of SCRIBES, including training, with per-page LLM inference in Figure 4. Let $g$ denote the number of groups processed. While per-page inference (dashed purple line) increases linearly with both the number of groups $g$ and the group size $k$, the SCRIBES-trained model yields substantial FLOP savings, with the magnitude of savings growing proportionally to group size. For instance, with 100 webpages per website group, SCRIBES can already provide a computational saving of $1.12 \times 10^{21}$ FLOPS when processing $10^5$ groups. Additional details on the FLOP estimates are provided in Appendix D.3.

---

> ### Author Response · Authors · 2025-11-25
>
> > Could you include one or two representative examples of the generated Python extraction scripts in the appendix , so that readers can better understand what the model actually produces?
>
> Thank you for this suggestion. We have included examples in our updated Appendix I.
>
>
> > Is it feasible to integrate a lightweight Open IE (or similar semantic post-processing) component into the generated script to improve its ability to handle less strictly structured content?
>
> Thank you for these insightful questions. As one potential direction of future work, an organic way of incorporating semantic post-processing could be to encourage the LLM to generate scripts that could leverage packages like spaCy / CoreNLP. We will leave them for further exploration.

---

### Official Review · Reviewer_oCsj · 2025-11-01

**Soundness:** 2
**Presentation:** 3
**Contribution:** 2
**Rating:** 4
**Confidence:** 4

**Summary:**

The paper addresses the task of knowledge extraction, i.e. extraction of triplets (subjects, predicates, and objects), from semi-structured web content such as tables and infoboxes. This paper proposes to extract such knowledge with LLM-generated extraction script that is designed to be applicable to similar pages from the same source website. Furthermore, the authors use reinforcement learning (RL) to train LLMs to produce these scripts. The results show that for the 14B and 32B models, this script generation approach notably outperforms direct LLM-based direct extraction, and the application of RL provides a subsequent performance gain.

**Strengths:**

1. Script-based knowledge extraction holds promise for significantly improving the speed and efficiency of the extraction process compared to direct, single-instance LLM extraction.
2. The application of RL brings a clear and significant improvement in performance compared to the base models.

**Weaknesses:**

1. While the extraction of knowledge triplets can ideally benefit downstream tasks such as knowledge-intensive question answering, I am not completely convinced that it is necessary. The knowledge triplet is a relatively narrow format, and may not be sufficient to comprehensively represent the knowledge. A promising alternative could be to directly use the source data through RAG.
2. The reported performance seems inconsistent. For example, the script generation method using the gpt-oss-120b model appears to significantly lag behind the results achieved by direct extraction with the same model.
3. The comparison against established baselines is not comprehensive enough. I would love see a more thorough comparison, including against the numbers previously reported for the SemiBench paper itself, and other related baselines such as specialized relation extraction models, LLM-based relation extraction agents, etc.
4. Could the authors present more details regarding the results in Section 5.1? Where does the QA data come from, and how is it selected?
5. I would also appreciate more implementation details about HTML flattening and cleaning, and whether different strategies were explored. These pre-processing steps can have a dramatic impact on both the performance and the resulting behavior of the extraction system.

**Questions:**

As in weakness

---

> ### Author Response · Authors · 2025-11-25
>
> We thank the reviewer for your insightful feedback for our paper! Below, we will address your concerns point-by-point. We have also updated the paper with new experiments highlighted in red and will also include results in our response below:
>
> > While the extraction of knowledge triplets can ideally benefit downstream tasks such as knowledge-intensive question answering, I am not completely convinced that it is necessary. The knowledge triplet is a relatively narrow format, and may not be sufficient to comprehensively represent the knowledge. A promising alternative could be to directly use the source data through RAG.
>
> Thank you for raising this important point. While we acknowledge that knowledge triplets may seem like a relatively narrow representation, we argue that they provide valuable structural signals for interpreting semi-structured data embedded in web HTMLs. As shown in Section 5.1 of our paper, incorporating these triplets as augmentation to flattened HTML leads to consistent QA improvements across models, including over 4% for GPT-4o. Rather than serving as an alternative to RAG, SCRIBES can be viewed as **a complementary approach that enhances RAG performance** in leveraging semi-structured web pages for question answering.
>
> > The reported performance seems inconsistent. For example, the script generation method using the gpt-oss-120b model appears to significantly lag behind the results achieved by direct extraction with the same model.
>
> Thank you for raising this clarification question. We believe that **generating scripts to parse knowledge is inherently more challenging than directly extracting triples from flattened HTMLs**, since the latter primarily requires identifying and copying relevant information, whereas the former **additionally** requires understanding similarities between webpages layouts within the same web domain and generating scripts to extract knowledge. This expectation is consistent with our empirical findings: as shown in Table 1, for all tested models (Q-14B, Q-32B, GO-20B, GO-120B, and L-70B), the best-performing script generation methods consistently lag behind the best direct LLM extraction baselines. Therefore, the reported performance is consistent with our hypothesis.
>
> > The comparison against established baselines is not comprehensive enough. I would love see a more thorough comparison, including against the numbers previously reported for the SemiBench paper itself, and other related baselines such as specialized relation extraction models, LLM-based relation extraction agents, etc.
>
> Thank you for raising this important point:
>
> 1.  We have updated Table 1 in the revised paper to include results reported in SemiBench. Notably, SemiBench primarily conducts experiments on the cleaned webpages portion of the dataset, whereas our work focuses on whole webpages, which present greater noise and structural variability. On this setting, SemiBench provides only limited results using direct LLM extraction and fine-tuned models. As shown in the updated table, these reported numbers are **consistently lower than those achieved by our best direct extraction baseline (GO-120B 2-shot flatten), and also lower than our best SCRIBES-trained model**. The relevant parts of the table are also shown below:
>
> | Model & Method | $R^{\mathrm{LM}}$ | $P^{\mathrm{LM}}$ | $F_1^{\mathrm{LM}}$ |
> |----------------|---------------------|----------------------|------------------------|
> | **Baselines (Direct LLM Extraction)** | | | |
> | *L-70B* (Reported by [1]) | 24.3 | 15.7 | 19.1 |
> | *Fine-tuned L-70B* (Reported by [1]) | 21.4 | 27.1 | 23.9 |
> | *GPT-4o* (Reported by [1]) | 35.1 | 23.8 | 28.3 |
> | Q-14B flatten | 30.5 | 36.5 | 29.9 |
> | Q-32B flatten | 28.7 | 37.4 | 29.9 |
> | GO-20B 2-shot flatten | 33.2 | **47.1** | 34.9 |
> | GO-120B 2-shot flatten | **42.3** | 46.3 | **40.4** |
> | **SCRIBES (Script-gen)** | | | |
> | Q-14B | 23.0 | 24.3 | 19.9 |
> | Q-14B (+ CC) | 25.2 | 23.0 | 21.8 |
> | Q-32B | 29.9 | 31.5 | 28.1 |
> | Q-32B (+ CC) | **37.4** | **36.0** | **33.2** |

---

> ### Author Response · Authors · 2025-11-25
>
> 2. In addition, we have conducted new experiments incorporating two SOTA systems for relation extraction and knowledge base construction: HippoRAG and AutoSchemKG. The corresponding results are presented in the updated Table 1, Table 10, and Appendix G.1. The relevant section is included below for reference:
>
> In addition to the simple 2-shot baseline, we profile two SOTA LLM-based agentic knowledge-base–construction baselines: HippoRAG [2] and AutoSchemaKG [3], representative of recent LLM-driven KG construction pipelines.
>
> HippoRAG is a retrieval-augmented generation framework that builds a knowledge graph as an embedding index, mimicking the role of the hippocampus in human memory. We use the first stage of their KG construction pipeline, which consists of two prompts, one for named entity recognition (NER) and one for triple extraction. We also replace their 1-shot example with the same 2-shot examples used in our baseline.
>
> AutoSchemaKG is a framework for web-scale KG construction over a pretraining-scale corpus. It calls three LLM modules on each webpage: (1) an entity–entity relationship extractor, (2) an entity–event relationship extractor, and (3) an event–event relationship extractor. These prompts are all zero-shot and are challenging to adapt, so we retain them as originally specified.
>
> As shown in Table below, the simple 2-shot baseline outperform both LLM-based baselines across all models evaluated on our task, including by more than 20% for the strongest model, GPT-OSS-120B. Moreover, they inherit the same cost inefficiencies, as each webpage requires multiple LLM calls.
>
> | Method | $R^{\mathrm{LM}}$ | $P^{\mathrm{LM}}$ | $F_1^{\mathrm{LM}}$ |
> |--------|----------------------|----------------------|------------------------|
> | **Baselines (Direct LLM Extraction)** | | | |
> | *Q-14B w/ AutoSchemaKG* | 2.1 | 8.26 | 8.17 |
> | *Q-14B w/ HippoRAG (2-shot)* | 8.49 | 32.24 | 16.12 |
> | Q-14B flatten | 30.46 | 36.46 | 29.87 |
> | *Q-32B w/ AutoSchemaKG* | 2.64 | 11.14 | 9.33 |
> | *Q-32B w/ HippoRAG (2-shot)* | 10.12 | 39.70 | 20.03 |
> | Q-32B flatten | 28.73 | 37.44 | 29.93 |
> | *GO-20B w/ AutoSchemaKG* | 5.57 | 11.96 |  9.26 |
> | *GO-20B w/ HippoRAG (2-shot)* | 8.26 | 23.06 | 14.02 |
> | GO-20B flatten | 36.94 | 37.88 |  33.61 |
> | GO-20B 2-shot flatten | 33.18 | 47.10 |  34.93 |
> | *GO-120B w/ AutoSchemaKG* | 6.52 | 16.97 | 12.28 |
> | *GO-120B w/ HippoRAG (2-shot)* | 28.57 | 12.12 |  17.22 |
> | GO-120B flatten | 36.43 | 34.59 | 31.74 |
> | GO-120B 2-shot flatten | 42.27 | 46.26 | 40.40 |
>
> > Could the authors present more details regarding the results in Section 5.1? Where does the QA data come from, and how is it selected?
>
> Thank you for this suggestion. We have included more details in Appendix H of our updated paper. Here is the relevant snippet:
>
> The QA pairs in Sec. 5.1 are collected by [1] through an LLM-generated followed by human-auditing process. We summarize their process below:
>
> First, a 70B Llama model generated initial question-answer pairs using webpage content and ground truth data. Then, these pairs were refined by:
> - Removing overly complex questions that required heavy reasoning, focusing instead on comprehension of semi-structured webpages.
> - Eliminating compound questions that combined multiple queries into one to avoid inflated difficulty.
> - Filtering out trivial questions that all tested models answered correctly, ensuring better differentiation among model performances.
>
> Finally, human auditors reviewed and removed any pairs that were ungrounded in the source content or contained incorrect answers. For more details and statistics, refer to [1].

---

> ### Author Response · Authors · 2025-11-25
>
> > I would also appreciate more implementation details about HTML flattening and cleaning, and whether different strategies were explored. These pre-processing steps can have a dramatic impact on both the performance and the resulting behavior of the extraction system.
>
> We have provided implementation details and detailed analysis in Appendix C of our original paper, where we also experimented with the simple strategy of whitespace removal. To recap:
>
> Raw HTMLs are often long and repetitive. We propose a simple and effective dedup algorithm to significantly cut down the token length of HTML pages while still maintaining its structure. Algorithm 1 shows the implementation of this algorithm. We set $z=3$ in our experiments.
> Table 5 shows the token saving effect of our dedup algorithm. Removing whitespaces in a HTML only brings minimal token savings ($<$ 2\%), while our dedup algorithm brings significant token savings, cutting down token usage from $>$114k to $<$17k. We also profiled performance gains of baselines models using dedup. As shown in Table 6, employing deduplicated HTML yields clear improvements compared to using raw HTML. Most notably, deduplication significantly increases the Non-Empty Rate of baseline performance by enabling more data points to fit within the model's context window.
>
> **Table 5. Token reduction analysis across collected webpages**
>
> | **Processing Stage**          | **Avg Tokens** | **Percentage** |
> |-------------------------------|----------------|----------------|
> | Original tokens               | 114,318.6      | 100.0%         |
> | After whitespace removal      | 112,279.0      | 98.2%          |
> | After dedup                   | 16,985.1       | 14.9%          |
> | **Reductions**                |                |                |
> | Whitespace token savings      | 2,039.6        | 1.8%           |
> | Total dedup token savings     | 97,333.5       | 85.1%          |
>
>
> **Table 6. Performance comparison of baseline models using raw or dedup-ed HTML**
>
> Here, we feed each page one-by-one into the model and evaluate performance only on that page. Non-Empty Rate is \(1\) if the model produced at least one triple for the page, and \(0\) otherwise.
>
> | **Model & Format**        | $P^{\mathrm{LM}}$ | $R^{\mathrm{LM}}$ | $F_1^{\mathrm{H, LM}}$ | Non-Empty Rate |
> |---------------------------|----------------------|----------------------|---------------------------|----------------|
> | L-70B w/ Raw HTML         | 3.4  | 3.7  | 3.5  | 37.9 |
> | L-70B w/ Dedup HTML       | 14.2 | 9.5  | 11.3 | 46.4 |
> | GPT-4o w/ Raw HTML        | 13.7 | 15.4 | 14.5 | 63.8 |
> | GPT-4o w/ Dedup HTML      | 19.1 | 23.0 | 20.9 | 94.9 |
>
>
> References:
>
> [1] Kai Sun, Yin Huang, Srishti Mehra, Mohammad Kachuee, Xilun Chen, Renjie Tao, Zhaojiang Lin, Andrea Jessee, Nirav Shah, Alex Betty, Yue Liu, Anuj Kumar, Wen tau Yih, and Xin Luna Dong. Knowledge extraction on semi-structured content: Does it remain relevant for question answering in the era of llms?, 2025. URL https://arxiv.org/abs/2509.25107.
>
> [2] Bernal Jimenez Gutierrez, Yiheng Shu, Yu Gu, Michihiro Yasunaga, and Yu Su. Hipporag: Neuro-biologically inspired long-term memory for large language models. In The Thirty-eighth Annual Conference on Neural Information Processing Systems, 2024. URL https://openreview.net/forum?id=hkujvAPVsg.
>
> [3] Jiaxin Bai, Wei Fan, Qi Hu, Qing Zong, Chunyang Li, Hong Ting Tsang, Hongyu Luo, Yauwai Yim, Haoyu Huang, Xiao Zhou, Feng Qin, Tianshi Zheng, Xi Peng, Xin Yao, Huiwen Yang, Leijie Wu, Yi Ji, Gong Zhang, Renhai Chen, and Yangqiu Song. Autoschemakg: Autonomous knowledge graph construction through dynamic schema induction from web-scale corpora, 2025. URL https://arxiv.org/abs/2505.23628.

---

### Meta-Review · Area_Chair_cEW5 · 2026-01-05

**Summary:**

The paper introduces SCRIBES, a framework designed for web-scale semi-structured data extraction. Unlike standard approaches that run heavy LLM inference per page, SCRIBES uses Reinforcement Learning (RL) to generate reusable Python scripts that can be applied to groups of structurally similar webpages.

**Reviewer Concerns:**

The reviewers initially recognized the practical significance of the problem and the novelty of the "script generation" paradigm for reducing computational costs. However, the initial scores (4, 6, 6, 4) reflected concerns regarding insufficient baselines, the absolute performance gap compared to direct per-page LLM extraction, generalization across domains, and the novelty of the algorithmic components.
The authors provided a robust rebuttal that included:
1) Comparison against SOTA agentic frameworks (HippoRAG, AutoSchemaKG) and SemiBench numbers, showing SCRIBES outperforms or matches them at a fraction of the cost.
2) A detailed FLOPS comparison (Figure 4) demonstrating the massive efficiency gains at scale ($10^{21}$ FLOPS saved).
3) An ablation study training on finance/legal and testing on products/encyclopedias to prove generalization.
4) Statistical evidence showing a strong correlation (0.957) between their proxy reward (Fuzzy F1) and LLM-based evaluation.

Given the significant efficiency advantages for web-scale extraction and the comprehensive new experiments addressing the weak baseline critique, the work is solid. Thus, the decision leans towards Acceptance.

**Reviewer Scores:**

The primary complaint of Reviewer oCsj is comparison against established baselines is not comprehensive enough. The authors directly addressed this with the HippoRAG/AutoSchemaKG comparisons. The authors corrected a factual error regarding the "proprietary models" (pointing out Qwen-32B is open source) and provided the requested quantitative correlation for the reward metric. These direct rebuttals typically mitigate the "Soundness" concerns that drive lower scores.

---

### Decision · Program_Chairs · 2026-01-26

Accept (Poster)